# ER-mitochondria contacts promote mtDNA nucleoids active transportation via mitochondrial dynamic tubulation

Jinshan Qin[1], Yuting Guo[2,3], Boxin Xue[1], Peng Shi [4], Yang Chen[4], Qian Peter Su[1,5], Huiwen Hao[1], Shujuan Zhao[1], Congying Wu[4], Li Yu[6], Dong Li [2,3✉] & Yujie Sun [1✉]

A human cell contains hundreds to thousands of mitochondrial DNA (mtDNA) packaged into nucleoids. Currently, the segregation and allocation of nucleoids are thought to be passively determined by mitochondrial fusion and division. Here we provide evidence, using live-cell super-resolution imaging, that nucleoids can be actively transported via KIF5B-driven mitochondrial dynamic tubulation (MDT) activities that predominantly occur at the ER-mitochondria contact sites (EMCS). We further demonstrate that a mitochondrial inner membrane protein complex MICOS links nucleoids to Miro1, a KIF5B receptor on mitochondria, at the EMCS. We show that such active transportation is a mechanism essential for the proper distribution of nucleoids in the peripheral zone of the cell. Together, our work identifies an active transportation mechanism of nucleoids, with EMCS serving as a key platform for the interplay of nucleoids, MICOS, Miro1, and KIF5B to coordinate nucleoids segregation and transportation.

[1] State Key Laboratory of Membrane Biology, Biomedical Pioneer Innovation Center (BIOPIC), School of Life Sciences, Peking University, 100871 Beijing, China. [2] National Laboratory of Biomacromolecules, CAS Center for Excellence in Biomacromolecules, Institute of Biophysics, Chinese Academy of Sciences, 100101 Beijing, China. [3] College of Life Sciences, University of Chinese Academy of Sciences, 100049 Beijing, China. [4] School of Basic Medical Sciences, Peking University, 100191 Beijing, China. [5] School of Biomedical Engineering, Faculty of Engineering and Information Technology, University of Technology, Sydney, NSW 2007, Australia. [6] School of Life Sciences, Tsinghua University, 100084 Beijing, China. ✉email: lidong@ibp.ac.cn; sun_yujie@pku.edu.cn

In humans, mitochondrial DNA (mtDNA) encodes 13 proteins, 2 ribosomal RNAs, and 22 transfer RNAs. Mutations in mtDNA and nuclear genes related to mtDNA maintenance have been found to cause mitochondrial dysfunction and various human diseases, such as neurodegenerative and senescence-linked disorders[1]. Every cell possesses multiple copies of mtDNA, and their replication, segregation, and distribution are essential for mtDNA maintenance, as well as mitochondrial functions[1–4]. mtDNA is packaged into nucleoids in mitochondria. In the current view, nucleoids are thought to be constrained at the mitochondrial inner membrane[5], and their segregation and distribution in cells are passively dependent on the fusion, division, and relocation of mitochondria[6–8]. This viewpoint is based mainly on the observations that: (1) blocking mitochondrial fusion results in a large fraction of mitochondria devoid of nucleoids[7], and (2) disruption of mitochondrial division causes enlarged and clustered nucleoids[8]. However, in a recent study, while aggregation and mislocalization of nucleoids upon depletion of the mitochondrial inner membrane-associated MICOS complex is supposedly caused by the reduction of Drp1 and mitochondrial division, the restoration of mitochondrial division by Drp1 overexpression is unable to fully rescue the proper distribution of nucleoids[9]. This indicates that the passive segregation and partition processes are not enough for proper nucleoids distribution in the cell. Moreover, in certain specialized cell types, such as skeletal muscle cells, mitochondria demonstrate highly constrained mobility and infrequent fusion, yet their nucleoids are still distributed evenly[10]. These lines of evidence suggest that there may be other unidentified mechanisms, perhaps active ones, that are essential for nucleoids segregation and distribution in mitochondria.

We previously described mitochondrial dynamic tubulation (MDT), an activity of mitochondria, as a critical mechanism for the formation of the mitochondrial network in the peripheral zones of mammalian cells[11]. Different from fusion and division, MDT is an active process in which thin tubules are rapidly extended and retracted from mitochondria, and it is driven by the motor protein KIF5B along the microtubule. Generation of these dynamic tubules by MDT and their subsequent fusion are required for the formation of mitochondrial network. Interestingly, it was reported that bacteria exchange materials with each other via membrane protrusions, which are dynamic tubule-like structures. Moreover, live-cell imaging demonstrates that in muscle cells, matrix-targeted green fluorescent protein (GFP) can be transferred from one mitochondrion to another via a transient dynamic tubular connection[12]. A previous study has shown that mtDNA is in close proximity to KIF5B and microtubules, suggesting that mtDNA may be linked directly or indirectly to the motor[5]. Thus, we wonder if KIF5B motor and its related MDT may transport nucleoids, functioning as an active mechanism for proper partition and distribution of nucleoids within the mitochondrial network.

Here, using multicolor, live-cell super-resolution imaging, we provide evidence for active transportation of nucleoids driven by MDT, and prove that such active transportation is an additional mechanism required for the proper distribution of nucleoids in the mitochondrial network. We observe MDT activities frequently occur at the EMCS, where mtDNA is often localized and synthesized[13]. Riding in the thin dynamic mitochondrial tubule generated by MDT, the nucleoids also undergo active movement toward the tip of the tubule, and subsequent fusion of these dynamic tubules leads to the transfer of nucleoids among mitochondria and within the mitochondrial network. We further demonstrate that Mic60 on the mitochondrial inner membrane links mitochondrial nucleoids to the mitochondrial outer membrane protein Miro1, as well as KIF5B at the EMCS. Downregulation of Mic60 significantly destabilizes the endoplasmic reticulum (ER)–mitochondria contacts, decreases the MDT activity and reduces the endogenous level of Miro1. Using a TET-ON assay to induce in situ reformation of mitochondrial network in the cell, we show that the loss of Mic60 results in repressed nucleoids motility and a large fraction of mitochondria devoid of nucleoids in the peripheral zone of the cell. Together, our work unveils an active mtDNA transportation mechanism based on a link spanning from mitochondrial inner matrix-localized mtDNA to motor protein KIF5B at the ER–mitochondria contact sites (EMCS), greatly advancing the understanding of mtDNA partition and distribution in the cell.

## Results

**MDT actively transports nucleoids.** To visualize the dynamics of MDT and nucleoids, we first co-transfected Cos-7 cells with the mitochondrial marker TOM20-GFP and a well-characterized total nucleoid population marker TFAM-mCherry, and then imaged with our recently developed grazing incidence structured illumination microscopy (GI-SIM)[14], a live-cell super-resolution technique capable of imaging intracellular dynamics over a long time course. We observed mitochondrial tubules rapidly extending from the tip or side of mitochondria, a process we previously named as MDT[11]. Interestingly, we found that most of the MDT initiation sites (72%, $n = 59$) were spatially linked to the mitochondrial nucleoids (within 1 μm; Supplementary Fig. 1 and Supplementary Movie 1), and these nucleoids were often found to move into the dynamic tubules and ride along with MDT (57%, $n = 42$; Fig. 1a, b, Supplementary Fig. 2a–c and Supplementary Movie 2), with a high correlation in both spatial trajectory and velocity ($R = 0.92$; Fig. 1c, d). GI-SIM fluorescence imaging indicated that all MDT tubules were ranged from 80 to 350 nm in diameter (the lower bound is likely limited by the spatial resolution of GI-SIM; Supplementary Fig. 2d). Among all MDT tubules, the diameter of those containing nucleoids was ranged from 140 to 350 nm (Supplementary Fig. 2e), which is reasonable because STED imaging has revealed that nucleoids have a defined, uniform size of ~100 nm in mammals[15]. Importantly, for MDT tubules in the size range of mitochondrial nanotunnels (40–200 nm in diameter), a type of thin mitochondrial tubular structure reported previously[10], only a small subset (22%, $n = 36$) contained nucleoids. In contrast, majority (70%, $n = 23$) of the MDT tubules thicker than 200 nm were found to contain nucleoids (Supplementary Fig. 2f). Furthermore, we also noticed that the tubule size could change dynamically during MDT with the presence of nucleoids. As exemplified in Supplementary Fig. 2b, we observed that the nucleoid was paused at the junction between the thick and thin parts of the tubule, and could not run into the thin part until it became thicker. These analyses indicated that nucleoids indeed have lower chances to be transported in thinner tubules. Intriguingly, we noticed that for the mitochondrial nucleoids that moved into the dynamic tubules, 88% of them ended up locating at the tip of the tubule (Fig. 1e, Supplementary Fig. 2a–c and Supplementary Movie 2). This observation suggests that motions of the mitochondrial nucleoid and the tubule are not fully coupled, despite their high correlation in the overall process. Theoretically, the nucleoid can reach the tubule tip either by its own active forward movement or by retraction of the tubule. By scrutinizing the motility of mitochondrial nucleoids in the dynamic tubules, we found that besides the cases of well-coupled motility as exemplified in Fig. 1a, c, there were some cases of unsynchronized motility with the mitochondrial nucleoids sometimes moving faster than the tubular tip as exemplified in Fig. 1f, g and Supplementary Movie 2. These data indicated that in addition to riding with the MDT process, the nucleoids could also move actively in the tubule. Corroboratingly, such active and

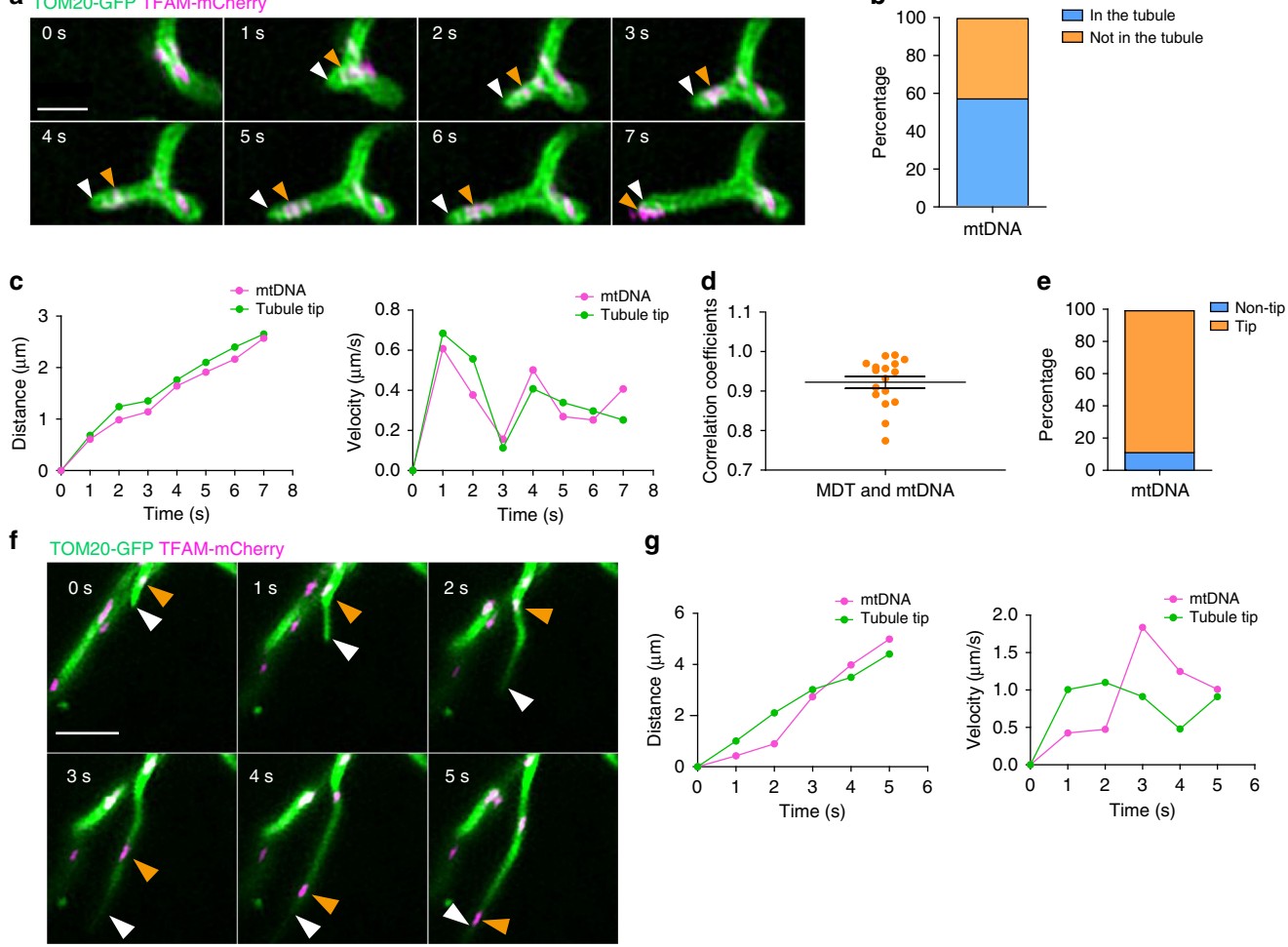

**Fig. 1 Mitochondrial DNA nucleoids are transported via MDT. a** Representative time-lapse images of MDT in a Cos-7 cell expressing Tom20-GFP and TFAM-mCherry demonstrate the transportation of TFAM-labeled nucleoids spatially linked to the MDT process. White arrowheads mark the tips of tubules generated by the MDT processes. Orange arrowheads indicate the sites of nucleoids. Additional examples are shown in Supplementary Fig. 2a–c. **b** Percentage of nucleoids near the MDT initiation sites that get transported into the thin tubule (blue box) or remain at the MDT initiation sites (orange box) for over 2 min, n = 42 events. **c** The trajectory and velocity of the nucleoid and that of the dynamic tubule during the tubulation process in **a**. **d** Spatial cross-correlation analysis of the tubulation process (Tom20-GFP) and nucleoid (TFAM-mCherry) trajectories, n = 17 events examined over three independent experiments. Data are shown as mean ± SEM. **e** Percentage of nucleoids in dynamic tubules transported to the tip of the tubules (orange box) or not the tip of the tubules (blue box) during the MDT process, n = 17 events. **f** Representative time-lapse images of MDT in a Cos-7 cell expressing Tom20-GFP and TFAM-mCherry demonstrate unsynchronized motility of TFAM-labeled nucleoids and the MDT process. White arrowheads mark the tips of tubules generated by the MDT processes. Orange arrowheads indicate the sites of nucleoids. **g** The trajectory and velocity of the mitochondrial DNA nucleoid and that of the dynamic tubule during the tubulation process in **f**. Scale bars: **a**, 1 μm and **f**, 2 μm. Source data are provided as a Source data file.

directional movement of mitochondrial nucleoids was also observed in the mitochondrial network ($\alpha = 1.54$, Supplementary Fig. 3 and Supplementary Movie 3).

It is well known that mitochondrial fusion and fission play important roles in the organization of nucleoids. As a result, we examined the effects of mitochondrial fusion and fission on the MDT-mediated active transportation of mitochondrial nucleoids. We blocked mitochondrial fission by depleting factor Drp1 in mouse embryonic fibroblast (MEF) cells using small interfering RNA (siRNA), and co-transfected these cells with TOM20-GFP and TFAM-mCherry (Supplementary Fig. 4b). In Drp1 RNA interference (RNAi) cells, we found that a small number of mitochondrial nucleoids were present in the form of enlarged clusters within aberrant and elongated mitochondria (Supplementary Fig. 4a), a morphological feature normally observed when Drp1 is depleted[8]. Nevertheless, in these Drp1-depleted cells, we were still able to observe frequent MDT events that could

actively transport the nucleoids (Supplementary Fig. 4c). Similarly, we could also observe frequent events of MDT-mediated nucleoids transportation in MEF cells downregulated with Mfn1 and Mfn2, factors responsible for mitochondrial fusion[16,17], despite that the mitochondria network was completely disrupted (Supplementary Fig. 4d–f). Therefore, the MDT-mediated transportation of mitochondrial nucleoids is independent on mitochondrial fission and fusion.

Taken together, these observations indicate that mitochondrial nucleoids can ride in the dynamic mitochondrial tubule generated by MDT and move toward the tip of the tubule, in striking contrast to the long-believed viewpoint that nucleoids mainly take constrained diffusive motion within the inner membrane of mitochondria. This prompted us to investigate the molecular basis of MDT-dependent active transportation of mitochondrial nucleoids, as well as its role in regulating nucleoids distribution in the mitochondrial network.

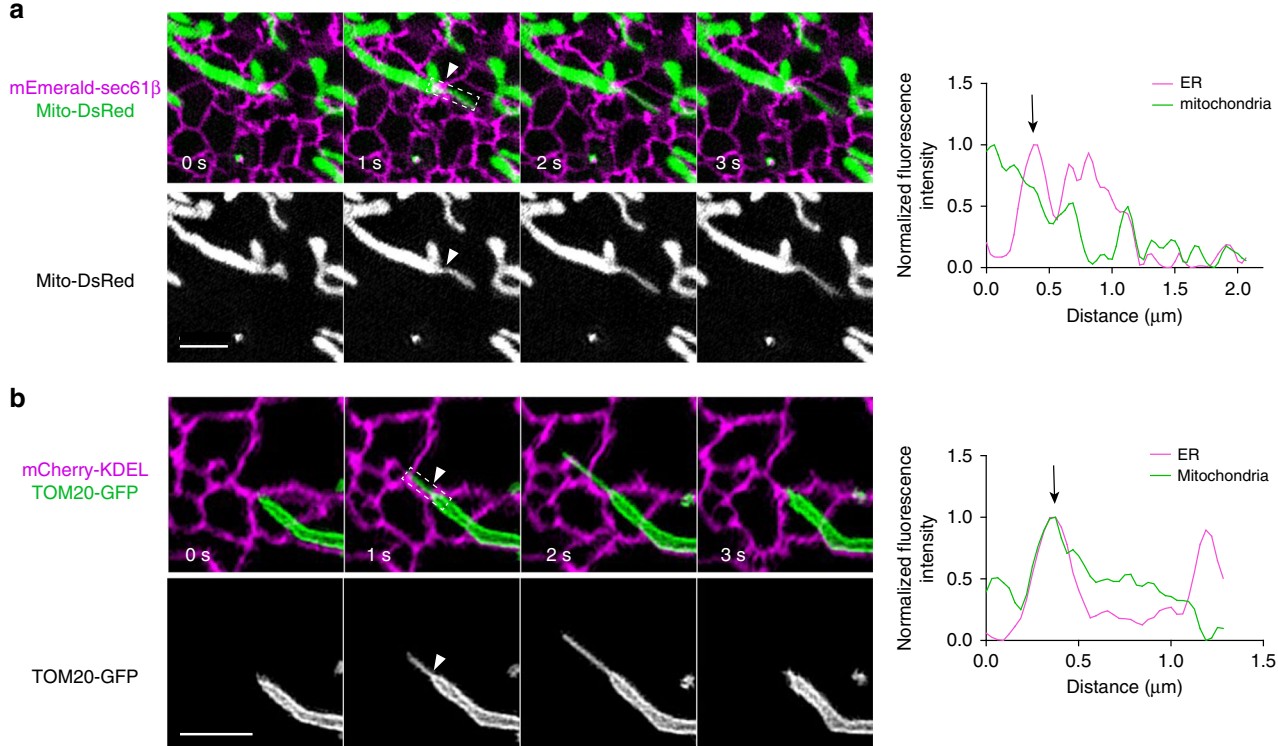

**Fig. 2 ER tubules mark the initiation sites of MDT. a, b** Examples of MDT at ER–mitochondria contacts. Left-hand images show Cos-7 cells expressing **a** mEmerald-Sec61β and mito-DsRed, and **b** mCherry-KDEL and Tom20-GFP, merged as indicated. Right graphs are linescans drawn through the mitochondria (dashed box), and show the relative fluorescence intensity of mitochondria (green) and ER (purple) along their lengths. White arrowheads at constrictions in the images correspond to the black arrows shown on the corresponding linescan graphs. Scale bars: 2 μm. Additional examples are shown in Supplementary Fig. 5a–d.

**ER tubules mark MDT initiation sites**. Lewis et al. have reported that in human cells, nucleoids are spatially linked to EMCS[13]. Given that report, in conjunction with our observations that a majority of MDT events initiated at sites with nucleoids nearby (Supplementary Fig. 1a–c), we were curious whether MDT is spatially correlated with EMCS. To check this, we co-transfected Cos-7 cells with an ER marker (mEmerald-Sec61β) and mito-DsRed, or mCherry-KDEL and TOM20-GFP, and used GI-SIM to image regions of the cell periphery where contacts between ER and mitochondria could be well resolved. Indeed, we observed that MDT events occurred predominantly at the sites of contact between ER and mitochondria (90%, n = 51 from 23 cells; Fig. 2a, b, Supplementary Fig. 5 and Supplementary Movie 4). There was no significant difference between thin MDT tubules (diameter ≤ 200 nm) and thick MDT tubules (diameter > 200 nm) initiating at the EMCS (Supplementary Fig. 5e). Two EMCS examples in which ER tubules crossed over mitochondria (white arrows) are given in Fig. 2a, b. At the EMCS, mitochondrial diameters decreased quickly, and a thin tubule extended from the mitochondria. The corresponding linescan analysis (Fig. 2a, b, right) revealed that MDT was spatially linked to EMCS. To further test the idea that MDT-mediated nucleoids transportation is initiated at EMCS, Cos-7 cells were transiently transfected with mCherry-KDEL, TOM20-GFP, and TFAM-HaloTag. The TFAM-HaloTag was then bound by a cell-permeable fluorescent dye, Janelia Fluor 646 (JF$_{646}$) (ref. [18]; Supplementary Fig. 6a and Supplementary Movie 5). We observed that nucleoids were generally adjacent to EMCS (Supplementary Fig. 6b), consistent with the previous report[13]. As shown in a representative time-lapse series, nucleoids at EMCS were indeed frequently transported by MDT in live Cos-7 cells (Supplementary Fig. 6c and Supplementary Movie 5). Thus, MDT occurred at positions where ER tubules contacted mitochondria, and it also mediated partition and transportation of nucleoids near the EMCS.

**Miro1 is enriched at EMCS and is required for MDT**. In our previous study, we showed that MDT is driven by KIF5B along the microtubule[11]. Interestingly, it was reported that KIF5B is recruited onto mitochondria by Miro1, a well-known outer mitochondrial membrane protein[19–23], and it was also reported that the homologous protein of Miro1 in yeast, Gem1, is localized at EMCS and regulates ER–mitochondria contacts[24]. To verify the spatial correlation between Miro1 and EMCS in mammalian cells, we transiently transfected Cos-7 cells with mEmerald-Sec61β and mito-DsRed, and used a monoclonal antibody directly against hMiro-1 to localize Miro1 using immunofluorescence. Miro1 staining, observed in a few foci per mitochondrion (Fig. 3a), frequently coincided with ER tubules (71%, n = 10 cells; Fig. 3b, c), indicating that Miro1 is enriched at EMCS. The enrichment of Miro1 at EMCS can lead to a higher local concentration of KIF5B and promote MDT initiation at EMCS. To test this reasoning, we knocked down Miro1 by siRNA and observed that the MDT activity was greatly repressed in Miro1 knockdown cells (Supplementary Fig. 7a), indicating that Miro1 is required for MDT, similar with KIF5B (Supplementary Fig. 7b). Moreover, a dramatic change of mitochondrial distribution from a cell peripheral network to perinuclear accumulation was observed in cells with Miro1 depletion (Supplementary Fig. 7c), a phenotype also found in KIF5B knockout cells (Supplementary Fig. 7d)[11].

Given both mtDNA and Miro1 are closely positioned to EMCS, respectively, we also examined the spatial correlation between mtDNA and Miro1. We used antibodies directly against mtDNA and Miro1 in Cos-7 cells, and imaged by GI-SIM. We

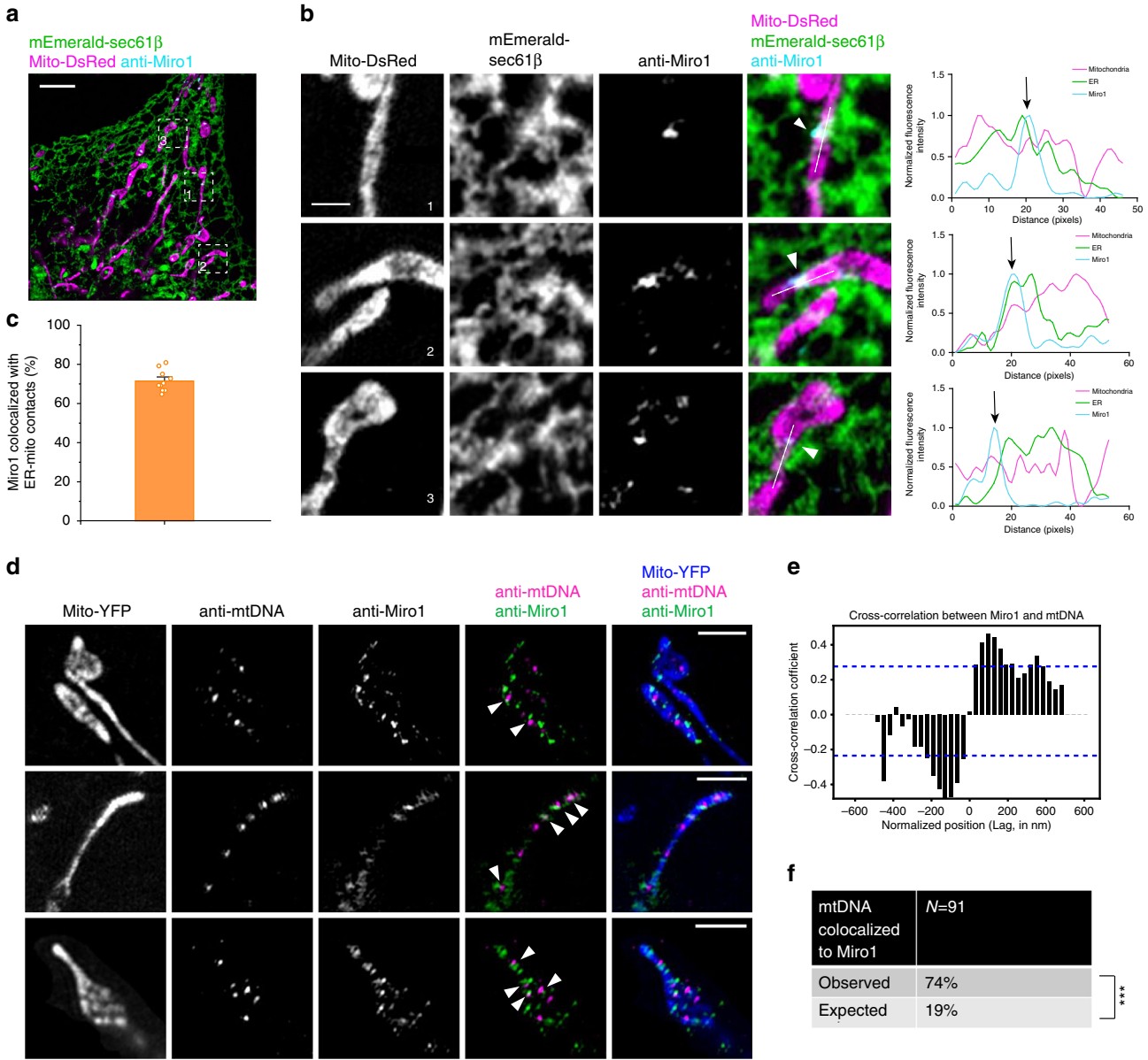

**Fig. 3 Miro1 is enriched at EMCS and spatially associated with mtDNA. a** A multicolor SIM image of a Cos-7 cell expressing mEmerald-Sec61β and mito-DsRed, with Miro1 labeled with α-Miro1 antibody. **b** Higher magnifications of the boxes in **a** showing three examples of Miro1 enriched at EMCS. White arrowheads indicate Miro1 punctae. Graphs to the right show relative pixel intensities of mito-DsRed, anti-Miro1, and mEmerald-Sec61β from a linescan drawn on each corresponding image (dashed line). The white arrowheads correspond to the black arrows on the linescans. **c** The percentage of Miro1 punctae that colocalized with the ER–mitochondria contacts in fixed Cos-7 cells. Data are shown as mean ± SEM ($n = 10$ cells examined over three independent experiments). **d** Cos-7 cells were transfected with mito-YFP, and immunostained by anti-Miro1 and anti-mtDNA. Arrowheads show mtDNA foci with adjacent Miro1 foci. **e** Spatial cross-correlation analysis of mtDNA and Miro1 fluorescence intensity along linescans of mitochondria ($n = 91$ mtDNA from 17 images) in fixed Cos-7 cells. Blue dotted lines indicate 95% confidence interval cutoffs, corresponding to the position of 224 nm. **f** Percentage of mtDNA that colocalized with Miro1 in Cos-7 cells (distance < 224 nm), corresponding to the 95% confidence interval marked by the blue dotted lines in **e**. Significantly more mtDNA is spatially (74%) associated with Miro1 than that expected by a randomly scrambled distribution (19%; $n = 91$ mtDNA from 17 images examined over three independent experiments, $P < 0.0001$, ***$P < 0.001$, two-tailed, unpaired Student's $t$-test). Scale bars: **a**, 5 μm; **b** and **d**, 2 μm. Source data are provided as a Source data file.

quantified the spatial relationship between Miro1 and mtDNA based on the Pearson correlation coefficient[13] and found that mtDNA signals were enriched within 224 nm laterally adjacent to Miro1 (Pearsons' $R = 0.46$; Fig. 3d, e). Among all mtDNA foci ($n = 91$), 74% demonstrated significantly higher spatial colocalization with Miro1 than that expected by coincidence (Fig. 3f and Supplementary Fig. 8), indicating that mtDNA is spatially associated with Miro1. As Miro1 is the receptor protein to recruit KIF5B onto mitochondria, these data are consistent with

an earlier work reporting that mtDNA is in close proximity to microtubules and KIF5B[5], supporting the implication that KIF5B may be involved in the active transportation of mtDNA in MDT.

**Mic60 links mtDNA to Miro1 and increases EMCS stability.** In yeast, MICOS, a multi-protein complex of the mitochondrial inner membrane, was reported to mediate the association between the mitochondrial nucleoids and the yeast Miro1 ortholog

Gem1 (ref. [25]). Moreover, Mic60, a main subunit of the MICOS complex in mammals, is reportedly found in purified HeLa mtDNA[26], and endogenous human Miro1 and Mic60 were found to be associated in the same complex[27,28]. Mic60 contains a transmembrane domain that spans the inner mitochondrial membrane with the bulk of the protein protruding into the intermembrane space[29]. These lines of evidence suggest that the MICOS complex may also mediate the association between nucleoids with Miro1 and KIF5B in mammalian cells. Indeed, immunoprecipitation of Mic60 in HEK293T cells revealed interaction with the main component of the mammalian nucleoids TFAM and with Miro1 (Fig. 4a). We then assessed the spatial links among Mic60, Miro1, and mtDNA by immunofluorescence. Both mtDNA (Fig. 4b and Supplementary Fig. 9a) and Miro1 (Fig. 4c and Supplementary Fig. 9b) were found to localize adjacent to Mic60 with high spatial correlation (Pearsons' $R = 0.6$ for mtDNA and 0.7 for Miro1). Interestingly, in addition to their spatial association, we also found that Mic60 knockdown in Cos-7 cells (Supplementary Fig. 9c) significantly decreased both MDT events (Supplementary Fig. 9d) and endogenous levels of Miro1 (Supplementary Fig. 9e). Together, these findings support a molecular linkage spanning from mitochondrial inner matrix-localized mtDNA to Mic60, Miro1, and KIF5B, which drives the active nucleoids transportation in mitochondria.

The decrease of endogenous levels of Miro1 upon Mic60 depletion (Supplementary Fig. 9e) suggests that the MICOS complex may contribute to the organization, and mutual interactions between ER and mitochondria. We imaged Mic60 RNAi and control Cos-7 cells expressing mEmerald-Sec61β and mito-DsRed, and quantified the colocalization level between ER and mitochondria based on the overlapping perimeter profiles of those organelles (Fig. 4d), as described previously[30]. The data showed that the percentage of mitochondrial perimeter colocalizing with ER was indistinguishable between Mic60 RNAi cells and control cells (Fig. 4e). Then, to examine whether the MICOS complex contributes to the stability of EMCS, we monitored the dynamics of ER–mitochondria contacts in cells by using time-lapse images taken every 30 s for 5 min (Fig. 4f, g). As shown in Fig. 4h, the overlapping pixel profiles obtained overtime from entire cells increased in Mic60 RNAi cells, compared to that in control cells, indicating that contact stability decreased in Mic60 RNAi cells.

**MDT contributes to nucleoids proper distribution.** We have previously proved that KIF5B drives MDT, which functions as an additional mechanism for formation of mitochondrial networks in the peripheral zones of a cell. With the finding that KIF5B also links to nucleoids, we were curious how MDT and nucleoids active transportation, both driven by KIF5B, are coordinated to regulate nucleoids distribution in the mitochondrial network. We first investigated the transportation and partition of nucleoids during the mitochondrial network formation process driven by MDT. To do so, we transiently transfected TFAM-mCherry into KIF5B$^{-/-}$ NRK cells that stably express mito-YFP and TET-ON-KIF5B. In KIF5B knockout cells, mitochondrial network disappeared and mitochondria accumulated around the perinuclear region (Fig. 5a, 0 h). Upon tetracycline induction to express KIF5B[11], we observed that thin, highly dynamic tubules were pulled out of the perinuclear mitochondria, and fusion of these dynamic tubules gradually restored the mitochondrial network in the peripheral zones of the cell. During this mitochondrial network reformation process, we found that the dynamic distribution of mitochondrial nucleoids in the peripheral mitochondrial network takes place in three stages (Fig. 5a). During stage 1 (0–1 h upon tetracycline induction), the nucleoids were actively

transported into the dynamic mitochondrial tubules that extended outward from the perinuclear mitochondria and subsequently ran to the tip of the tubules (Fig. 5b). In stage 2 (2–4 h upon tetracycline induction), the mitochondrial nucleoids at the tubule tips were handed over to other mitochondria upon fusion of these dynamic tubules (Fig. 5c, d). In stage 3, the peripheral mitochondrial networks were reformed, in which nucleoids were evenly distributed (after 4 h upon tetracycline induction).

Next, we set to investigate the role of KIF5B-based nucleoids motility in the dynamic distribution of nucleoids by depleting Mic60 in the cell, which is supposed to specifically disrupt the link between nucleoids and KIF5B. When knocking down Mic60 in Cos-7 cells, we found that the motility of nucleoids was significantly decreased and restricted (Fig. 5e and Supplementary Movie 6), causing nucleoids near the MDT initiation sites less likely to be transported into the dynamic tubules (Fig. 5f, g). Consequently, while inducing KIF5B expression with tetracycline was still able to restore the mitochondrial network in KIF5B$^{-/-}$ NRK cells with Mic60 depletion as in control cells (Fig. 5h, i), the nucleoids could not be transported to the newly restored mitochondrial network, but remained in the perinuclear region (Fig, 5h), leading to a much lower nucleoids density in the outer regions of the cell than that in the perinuclear region (Fig. 5j). Meanwhile, in the perinuclear region, the nucleoids became significantly enlarged (diameter > 1 μm) and the mitochondria containing such enlarged nucleoids became rounded, leaving many mitochondria devoid of nucleoids (Fig. 5h). These features are likely attributed to deficient mitochondrial fission, consistent with a previous report[9]. To confirm that the effects caused by Mic60 depletion is not cell-type specific, we also examined nucleoids distribution in Mic60 RNAi Cos-7 cells. We observed phenomena similar to those observed in NRK cells (Supplementary Fig. 10). Taken together, these data show that the MICOS complex plays dual roles in regulating nucleoids. On the one hand, it affects the segregation of nucleoids by reducing the levels of Drp1 and thus the fission of mitochondria[9]. On the other hand, through its interaction with nucleoids and Miro1, as well as KIF5B, the MICOS complex on the mitochondrial inner membrane is essential for the MDT-mediated nucleoids transportation, required for the proper distribution of nucleoids in the peripheral zone of a cell.

## Discussion

In this study, we identify an active partition and transportation mechanism of mitochondrial nucleoids via MDT at EMCS. In our previous work, we have shown that MDT is driven by KIF5B motility along microtubules and that fusion of dynamic mitochondrial tubules leads to mitochondrial network formation in the peripheral zone of the cell[11]. These features make MDT a particularly efficient way to regulate nucleoids allocation in the mitochondrial network. Based on colocalization analysis and loss-of-function assays, we have probed the molecular basis of MDT-mediated nucleoids partitioning and transportation. In this model, nucleoids near EMCS interact with the mitochondrial inner membrane MICOS complex. The MICOS complex is linked to the outer membrane protein Miro1, which is enriched at the EMCS and recruits KIF5B motors onto the mitochondria. Consequently, concentrated KIF5B makes an EMCS a hot spot for MDT activity, and nucleoids can be transported along with the thin tubules, both as hitchhikers and as self-movers (Fig. 6).

Previous studies have shown that EMCS coordinate mtDNA replication and mark the division sites where nascent mtDNA segregates to daughter mitochondria[13]. In this work, we found that MDT predominantly occurs at EMCS, where Miro1 foci and nucleoids are frequently colocalized. Therefore, we consider the

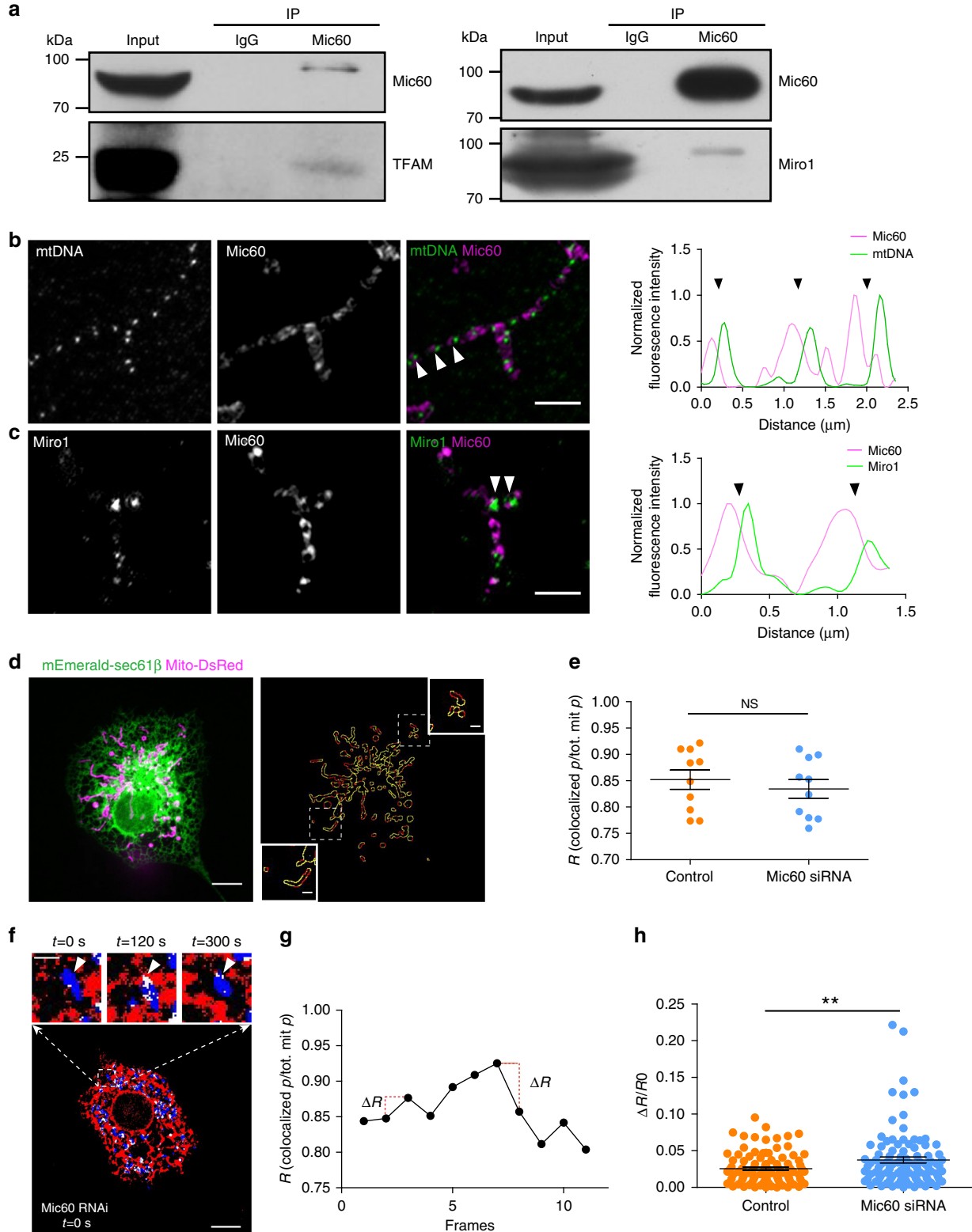

EMCS to be a key platform for mitochondrial nucleoids, MICOS, Miro1, and KIF5B to coordinate MDT and mtDNA segregation and transportation. Nevertheless, ER and mitochondria, as network structures found throughout the whole cell, actually form numerous contact sites that are either transient or stable at any given moment and, importantly, to ensure functionality among these EMCS, the cell must establish stable contacts. Interestingly, we found that Mic60 knockdown decreased EMCS stability

(Fig. 4d–h). We supposed that loss of Mic60 reduces Miro1 at the EMCS (Supplementary Fig. 9e) and Miro1 may interact with Mfn2 (ref. [31]), which is a potential tether between ER and mitochondria in mammalian cells[32,33].

The MDT-based mtDNA partition and transportation mechanism is distinct from mitochondrial fusion or fission. Blocking mitochondrial fusion results in a large fraction of mitochondria devoid of nucleoids[7], and disruption of mitochondrial division

**Fig. 4 Mic60 links mtDNA to Miro1 and contributes to EMCS stability. a** Co-immunoprecipitation of Mic60 with endogenous TFAM and Miro1 in HEK293T cells, and the immunoprecipitates were blotted as indicated. Input: 2%. **b** Cos-7 cells immunostained by anti-mtDNA and anti-Mic60. Arrowheads show mtDNA foci with adjacent Mic60 foci. (Right) the graph is linescan drawn through the mitochondria, and show the relative fluorescence intensity of mtDNA (green) and Mic60 (purple) along their length. White arrow positions correspond to black arrows on the linescan. **c** As in **b** for cells immunostained by anti-Miro1 (green on linescan) and anti-Mic60 (purple on linescan). **d** Dual-color confocal image of a Cos-7 cell expressing mito-DsRed and mEmerald-Sec61β (left), and the corresponding mitochondrial perimeter profile (right). Mitochondrial perimeter profiles are presented by red pixels, and the contacts between the mitochondrial perimeter and the ER are presented by yellow pixels. **e** ER–mitochondria colocalization index ($R$), calculated as the ratio between the perimeter pixels of mitochondria colocalized with the ER and the total perimeter pixels of mitochondria for the control and Mic60 RNAi cells ($n = 10$ cells were used per condition from three independent experiments). **f** Fluorescence image (modified as described in "Methods" section) of a Mic60 RNAi cell expressing mito-DsRed and mEmerald-Sec61β. The boxed region is enlarged and shows time-lapse images. White arrowheads indicate unstable ER–mitochondria interactions. **g** Representative kinetics of the ratio ($R$) on the entire Cos-7 cell shown in **f**. $\Delta R$ represents the difference between the $R$ of one image (frame) and that of the previous one. **h** Mean $\Delta R/R0$ values (where $R0$ is the $R$ of the first frame of each pair; $n = 110$ frames from ten control and ten Mic60 RNAi cells examined over three independent experiments). Scale bars: **b**, **c**, inset in **d**, and inset in **f**, 2 μm; **d** and **f**, 10 μm. Data are presented as mean ± SEM. $P$ value from top to bottom: $p = 0.4981$, $p = 0.0089$. NS no significant difference; **$p < 0.01$, two-tailed, unpaired Student's $t$-test. Source data are provided as a Source data file.

causes enlarged and clustered nucleoids[8]. Different from the effects on nucleoids caused by malfunction in fusion or fission, inhibiting MDT-mediated active nucleoids transportation results in significant loss of nucleoids within the mitochondrial network in the peripheral zone of the cell (Fig. 5j), an area so wide that cells can benefit from active transportation of mtDNA. Moreover, a recent study demonstrated that depletion of Mic60, the key molecule for MDT-based nucleoid motility identified in our study, resulted in 60.3% of MEF cells showing nucleoids aggregation, compared to 19.3% of Drp1-knockdown MEF cells[9]. Together with its effects in MDT, our data suggest that the MICOS complex is a regulator of nucleoid distribution, which functions in both nucleoid partition and transportation. Knockdown of Mic60 has also been reported to result in the loss of mitochondrial cristae junctions[34]. We speculate that the cristae disruption may decrease the possibility of nucleoids tethering to the inner membrane, and thus hamper its association with Miro1 and KIF5B for active transportation.

In the meantime, while the MDT-mediated transportation of mitochondrial nucleoids is independent on mitochondrial fission and fusion (Supplementary Fig. 4), these three mitochondrial processes may coordinate to properly distribute nucleoids in the mitochondrial network. As segregation of duplicated mtDNA is coupled with mitochondrial division[13], blocking mitochondrial division can result in abnormally enlarged nucleoids, which are possibly of hampered transportation in the thin MDT tubules. When mitochondrial fusion is blocked in the cell, the mitochondria network is completely disrupted, and mitochondria turn round or oval-shaped vesicles. Under such circumstances, even though MDT can still normally occur, the nucleoids transported by MDT would not be possible to exchange among different mitochondria without mitochondrial fusion, thus leading to a large fraction of mitochondria devoid of nucleoids (Supplementary Fig. 4). Therefore, mitochondrial fission, dynamic tubulation, and fusion may act in turn to facilitate the nucleoids proper distribution in the mitochondrial network.

The active mtDNA transportation model is especially crucial for cells with restricted mitochondrial motility, such as skeletal and cardiac muscle cells. Indeed, dynamic mitochondrial tubules have been observed in cardiac[12] and skeletal muscle cells[35], meaning that nucleoids may be exchanged between nonadjacent mitochondria via MDT activity. Importantly, a higher abundance of MDT is observed in the presence of mtDNA mutations in human cells[10,36], suggesting that MDT-based nucleoids partitioning and transportation may help mitochondria adapt to genetic stress. Inhibiting MDT-mediated active nucleoids transportation results in significant loss of nucleoids in the peripheral mitochondrial network. These nucleoids-devoid mitochondria are deficient of essential respiratory subunits encoded by mtDNA, which may cause defects in mitochondrial respiratory capacity. There is evidence suggesting that early dysfunction of mitochondrial respiration plays an essential role in PD pathogenesis[37]. Besides, previous studies have shown that the severity of this heterogeneous mtDNA loss corresponds well with the severity of mitochondrial membrane potential loss[6,38]. In addition, the peripheral mitochondria form contacts with the ER tubules, lysosomes, endosomes, peroxisomes, lipid droplets, and plasma membrane[39] for specific functions, such as exchange of lipids, ions, and metabolites between organelles. Devoid of nucleoids in these peripheral mitochondria may affect the interactions. Therefore, these findings would have broad implications for understanding segregation and transmission of mtDNA in disease and aging.

## Methods

**Plasmid construction.** Mito-DsRed plasmid, a gift from Dr. Quan Chen (Institute of Zoology, CAS), was constructed by fusing the mitochondrial target sequence of human TXN2 with the DsRed sequence. TOM20-GFP and mito-YFP, gifts from Dr. Li Yu (Tsinghua University)[11], were obtained by inserting the rat TOM20 coding sequence into pEGFP-N1 and by fusing the mitochondrial target sequence of human COX8A with the YFP sequence, respectively. We constructed the mito-BFP plasmid by replacing the DsRed of mito-DsRed with the BFP sequence. We also constructed TFAM-GFP by inserting the human TFAM coding sequence into pEGFP-N1 and TFAM-mCherry by replacing the GFP of TFAM-GFP with the mCherry sequence. TFAM-HaloTag was a gift from Dr. Dong Li (Institute of Biophysics, CAS) and mCherry-KDEL was a gift from Dr. Jianguo Chen (Peking University). We purchased mEmerald-Sec61β from Addgene (#90992). The primers used in this study are listed in Supplementary Table 1.

**Cell culture and plasmid DNA transfection.** Cos-7, NRK, MEF, and HEK293T, obtained from the American Type Culture Collection, were cultured in Dulbecco's modified eagle medium (Life Technologies) supplemented with 10% fetal bovine serum, 5% $CO_2$, and 1% each penicillin/streptomycin. Generation of normal rat kidney cells with a stable knockout of KIF5B or with tetracycline-inducible expression of KIF5B were described previously[11]. For GI-SIM imaging, coverslips and chambers were precoated with 5 μg/cm² fibronectin (Sigma), and cells were seeded to achieve ~60–80% confluency at the time of imaging. For confocal imaging, cells were seeded to 35 mm glass-bottom microscope dishes (Cellvis). Sixteen to 36 h prior to imaging, we transfected cells with 2 μg plasmid DNA in Opti-MEM medium (Invitrogen) containing 5 μl Lipofectamine 2000 reagent (Invitrogen), according to the manufacturer's instructions. Where indicated, the cells transfected with HaloTag plasmids were labeled with the JF$_{646}$ ligand (a gift from Dr. Luke D. Lavis, Janelia Research Campus), following the published protocol[18]. The cells were imaged immediately afterward.

**RNAi transfection and western blots.** For Drp1, Mic60, and Miro1 silencing, we used Drp1 siRNA (Ribobio, China), Stealth Mic60 siRNA (MSS293683, Life Technologies), Rhot1 siRNA (s30648, Invitrogen), and Silencer Negative Control N#1 siRNA (Ambion, Life Technologies). Cells were seeded in a 35 mm dish ~16 h prior to the first round of transfection. The cells were first transfected with 25 pmol RNAi oligonucleotides and 25 pmol negative control siRNA using Lipofectamine RNAiMax reagent (ThermoFisher Scientific), according to the manufacturer's instructions. Then, 48 h after the first round of transfection, cells were transfected again, this time with plasmids. After 12 h of plasmid DNA transfection, cells were replated onto 25 mm coverslips or 35 mm glass-bottom microscope dishes and

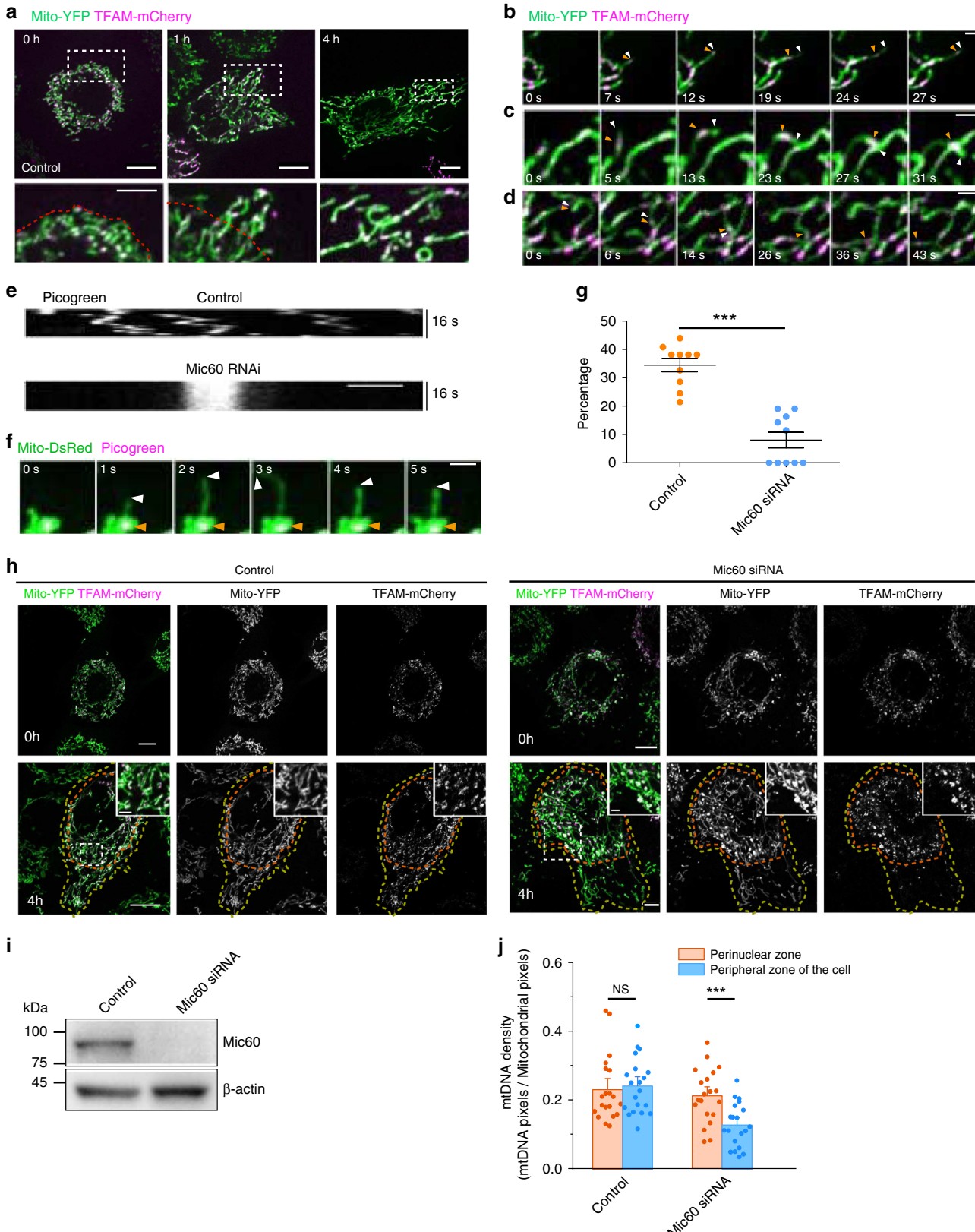

left overnight. Where indicated, cells were stained with PicoGreen DNA stain (1:1000 dilution, ThermoFisher) for 15 min at 37 °C before imaging.

For western blotting, whole cell lysates were resuspended in 2× SDS loading buffer, boiled for 10 mins, cleared by centrifugation, separated by SDS–PAGE, and transferred to a polyvinylidene difluoride membrane. We determined protein concentrations by doing gray value analysis on the western blot protein bands. Primary antibodies were used as follows: anti-Miro1, mouse monoclonal (WH0055288M1; Sigma-Aldrich), 1:200 dilution; anti-Mitofilin, rabbit polyclonal (10179-1-AP, Proteintech), 1:1000 dilution; anti-Mitofilin, mouse monoclonal (ab110329, Abcam), 1:500 dilution; anti-KIF5B, rabbit monoclonal (ab167429, Abcam), 1:1000 dilution; anti-Mitofusin1, rabbit monoclonal (ab57602, Abcam), 1:1000 dilution; anti-Mitofusin2, rabbit monoclonal (ab124773, Abcam), 1:1000 dilution; anti-DRP1, rabbit monoclonal (ab184247, Abcam), 1:1000 dilution; and anti-β actin, mouse monoclonal (sc-47778, Santa Cruz Biotechnology), 1:500

**Fig. 5 MDT contributes to nucleoids distribution in peripheral mitochondrial network. a** Confocal images of KIF5B$^{-/-}$ NRK cells expressing mito-YFP, TFAM-mCherry, and TET-ON-KIF5B and treated with 0.5 μg/ml tetracycline. Higher magnifications of the white dashed boxes are in the lower panels. Red dashed lines indicate the mitochondrial network boundary before tubulation. **b–d** KIF5B$^{-/-}$ NRK cells expressing mito-YFP, TFAM-mCherry, and TET-ON-KIF5B were treated with 0.5 μg/ml tetracycline for 1 h (**b**, **c**) and 2 h (**d**). White arrows indicate a dynamic tubule. Orange arrowheads indicate nucleoids. **e** Kymographs of mtDNA labeled by DNA dye picogreen in a control cell and a Mic60 RNAi cell. **f** Time-lapse images of Mic60 RNAi cells expressing mito-DsRed and stained by picogreen. White arrowheads indicate MDT processes. Orange arrowheads mark the sites of mtDNA. **g** Percentage of MDT events that drive mtDNA transportation to total MDT events in control and in Mic60 RNAi cells over a 5-min course (n = 10 cells were used per condition from three independent experiments). **h** KIF5B$^{-/-}$ NRK cells expressing mito-YFP, TFAM-mCherry, and TET-ON-KIF5B contain control or Mic60 siRNA. Both control and Mic60 RNAi cells were treated with 0.5 μg/ml tetracycline for 4 h. Yellow dashed lines indicate the boundary of the mitochondrial network. Red dashed lines indicate the boundary of the perinuclear mitochondria. Inset regions are magnified from the boxed areas. **i** Western blots of Mic60 indicate depletion of Mic60 in lysates of cells transfected with siRNA against Mic60. **j** Quantification of nucleoids density in the perinuclear zone and peripheral zone of control or Mic60 RNAi cells after adding 0.5 μg/ml tetracycline for 4 h (n = 20 cells were used per condition from three independent experiments). Scale bars: **a**, **h**, 10 μm; inset in **a**, 5 μm; **b–d**, **f** and inset in **h**, 2 μm; **e**, 1 μm. Data are presented as mean ± SEM. P value from top to bottom and left to right: p < 0.0001, p = 0.7101, p = 0.0006. NS no significant difference; ***p < 0.001, two-tailed, unpaired Student's t-test. Source data are provided as a Source data file.

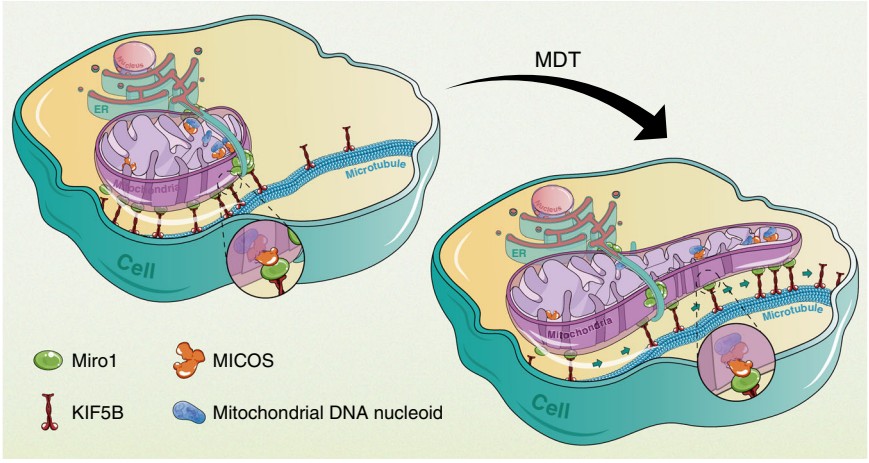

**Fig. 6 The role of MDT in the partitioning of mitochondrial DNA nucleoids.** At the ER–mitochondria contact sites, nucleoids interact with the inner membrane protein complex MICOS. MICOS is associated with the outer membrane protein Miro1, which is concentrated at the ER–mitochondria contact sites and recruits KIF5B motors on the mitochondria. When KIF5B-driven dynamic tubulation happens, nucleoids are transported via the interaction axis of nucleoid/MICOS/Miro1/KIF5B.

dilution. We also used horseradish peroxidase-conjugated goat anti-rabbit (7074 S, Cell Signaling Technology, 1:5000 dilution) or goat anti-mouse secondary antibodies (31436, Invitrogen, 1:5000 dilution) and the signal was detected with SuperSignal West Pico PLUS Chemiluminescent Substrate (ThermoFisher Scientific). Image lab (version 5.0) was used for western blotting images collection. Uncropped blots can be found in the Source data file.

**Immunoprecipitation**. Cells were lysed in lysis buffer (25 mM Tris-HCl pH 7.4; 1% NP-40; 150 mM NaCl; 1 mM EDTA; and 5% glycerol) containing protease inhibitors (Sigma-Aldrich). The lysed samples were incubated with mouse anti-Mic60 (ab110329; Abcam) or normal mouse IgG (A7028; Beyotime). After overnight incubation with gentle mixing at 4 °C, protein A + G agarose bead (P2012; Beyotime) was added to each sample and mixed for 3 h at 4 °C. Immunoprecipitates were washed extensively in lysis buffer, and bound complexes were separated by SDS–PAGE and transferred to a polyvinylidene difluoride membrane. For immunodetection, mouse anti-Mic60 (ab110329; Abcam) and mouse anti-Miro1 (WH0055288M1; Sigma-Aldrich) were used at 1:1000; rabbit anti-TFAM (HPA040648; Sigma-Aldrich) was used at 1:100.

**Immunofluorescence**. Cos-7 cells were seeded on preincubated coverslips in 35-mm dishes and transfected with plasmids, as previously described. The next day, we washed slides in PBS, fixed them in 6% paraformaldehyde in PHEM buffer (60 mM pipes (pH 6.9), 25 mM Hepes, 10 mM EGTA, and 2 mM MgCl$_2$) for 15 min at 20 °C, and washed the seeded coverslips three times with PHEM buffer, before incubating them for 30 min at room temperature with 10% BSA (Jackson, #001-000-162) and 0.5% TritonX-100 (ThermoFisher Scientific) in PHEM buffer. Next, we incubated the cells with primary antibodies (anti-Miro1, mouse monoclonal (WH0055288M1; Sigma-Aldrich), 1:2000 dilution; anti-Miro1, rabbit polyclonal (PA5-72835; Invitrogen); anti-DNA, mouse monoclonal (AC-30-10, Progen), 1:200 dilution; and anti-Mitofilin, rabbit polyclonal (10179-1-AP, Proteintech), 1:50 dilution) in BSA with 0.05% TritonX-100 for 1–2 h. The cells were

then washed with PHEM buffer three times, 5 min each wash, before being incubated for 60 min (protected from light) at room temperature with appropriately diluted, dye-labeled secondary antibodies. The slides were washed with PHEM buffer three times for 5 min each wash.

**Grazing incidence structured illumination microscopy**. Using a home-built structured illumination microscope, we took super-resolution images[14]. GI-SIM, where the illumination entering the objective rear pupil is launched just inside the critical angle for TIRF, creating an illumination field parallel to the substrate, is capable of imaging dynamic events near the basal cell cortex at 97-nm resolution and 266 frames/s over thousands of time points. Briefly, GI-SIM imaging was carried out using a high numerical aperture (NA) objective (Olympus APON 100XHOTIRF 1.7 NA), and detected using a sCMOS camera (Hamamatsu, Orca Flash 4.0 v3). Labview (version 2016) were used for image acquisition.

**Spinning disk confocal microscopy**. Conventional images were recorded by a spinning disk confocal microscope (PerkinElmer, UltraVIEW VoX, USA) equipped with a 100× 1.4 NA oil objective and an Ultra-888 electron multiplying charge-coupled device (Andor Technology). The live-cell incubation system (Chamlide 567 TC-W) was purchased from LCI, Korea. Volocity (version 6.3) were used for image acquisition.

**Analysis of the dynamics of MDT and mitochondrial nucleoids**. To visualize nucleoids transportation relative to MDT, Cos-7 cells expressing markers for the mitochondria and mitochondrial nucleoids were imaged every 1 s for up to 2 min in a single focal plane. We used the ImageJ plugin Manual Tracking to track MDT and mtDNA trajectories, and performed cross-correlation analyses between MDT and mtDNA movements by using the corrcoef function in Matlab (Mathworks). MSD fitting was performed with Origin analysis software (OriginLab).

**Colocalization analysis**. To analyze fluorescence colocalization, we performed a cross-correlation analysis between mtDNA and Miro1, mtDNA and Mic60, as well as between Miro1 and Mic60 using in-house written Python (version 2.7) script[13]. Firstly, we determine signal intensity along linescan of mitochondria. Then, correlations between two signal intensities along linescans of mitochondria were calculated. Next, correlations were recalculated as one of the signals was iteratively shifted in 32 nm increments and averaged. This analysis provided a distance threshold for colocalization.

Using the distance threshold for colocalization, we measured colocalization frequency in the merged images. To estimate the level of colocalization of randomly distributed Miro1 and mtDNA, we took the localization of Miro1 and of mtDNA from the experimental data, and randomized their localization using in-house written Matlab (version R2018a) script.

**Kymographs**. Kymographs of time-lapse images were produced by manually drawing lines across the images produced in the ImageJ plugin Multi Kymograph. The resulting graph represented the intensity against time along the line. The *x*-axis is the mtDNA position and the *y*-axis corresponds to time (moving from top to bottom).

**Mitochondrial and ER morphology analysis**. Mitochondrial and ER morphology analysis were calculated using the ImageJ plugins[30]. Original images were first converted to binary images after background subtraction in ImageJ. Next, we applied the ImageJ plugin IsoPhotContour2 to the binary images to obtain a profile of the mitochondrial perimeter, and then we merged that perimeter with the binary images of the ER. The ImageJ plugin Colocalization Highlighter was used to obtain ER–mitochondria colocalization pixels. The colocalization ratio *R* was calculated as the ratio between the pixel number of ER–mitochondria colocalization to the pixel number of the whole mitochondrial perimeter.

**Statistics and reproducibility**. Statistical significance between two values was determined using a two-tailed, unpaired Student's *t*-test (Graphpad Prism and OriginPro). All data are presented as the mean ± SEM; *$p < 0.05$; **$p < 0.01$; ***$p < 0.001$; NS, no significant difference. Statistical significances and sample sizes in all graphs are indicated in the corresponding figure legends. Each experiment was repeated three times independently with similar results. All images shown are representative results from biological replicates.

**Reporting summary**. Further information on research design is available in the Nature Research Reporting Summary linked to this article.

## Data availability

The original tif files of all images included in figures have been deposited in Zenodo with the identifier https://doi.org/10.5281/zenodo.3939035. All other data supporting the findings of this study are available within the article and its Supplementary Information files or from the corresponding authors upon reasonable request. Source data are provided with this paper.

## Code availability

All code used in this study have been deposited in Zenodo with the identifier: https://doi.org/10.5281/zenodo.3939035.

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

## Acknowledgements

We thank Prof. Jianguo Chen (Peking University) for mCherry-KDEL plasmid. We thank Dr. Luke D. Lavis (Janelia Research Campus) for JF$_{646}$ ligand. We thank the Core Facilities of the School of Life Sciences at Peking University for the confocal imaging support. We thank the Guangzhou Sagene Biotech Co., Ltd. for illustration support. This work is supported by funding from the National Key R&D Program of China, No. 2017YFA0505300, and the National Science Foundation of China 21825401 for Y.S; the National Science Foundation of China No. 91754202 for D. Li; China Postdoctoral Science Foundation (BX20190355) for Y.G; the National Key R&D Program of China 2017YFA0506500 for C.W.

## Author contributions

Y.S. and J.Q. planned the project. J.Q. and Y.G. performed the imaging experiments. Y.S., J.Q., B.X., and Q.P.S. analyzed the data. J.Q. carried out the plasmid construction and western blotting with the help of Y.C., H.H., and S.Z. P.S. performed the immunoprecipitation experiments. Y.S., D.L., and J.Q. wrote the manuscript with comments from L.Y. and C.W. All authors participated in the discussion of the manuscript.

## Competing interests

The authors declare no competing interests.
