## [Peer Review File · Nature Communications]

Peer Review Information

Manuscript title: ER-mitochondria contacts promote mtDNA nucleoids active transportation *via* mitochondrial dynamic tubulation

Corresponding author name(s): Prof Yujie Sun

Editorial notes:

none

Reviewer comments & decisions:

Reviewer comments, first version:

Reviewer #1 (Remarks to the Author):

Qin and colleagues developed a home-made super-resolution microscope, GI-SIM and applied it into the study of mitochondrial DNA nucleoids transportation. They described that the ER-mitochondria contact site and the MICOS complex are important in this process. Generally speaking, the study was well designed, performed and controlled. The major technology here is very unique and provides clear and convincing images. I will support further consideration of this manuscript for publication in Nature communication.

Listed below are a few points that will help improve the manuscript.

1) Overall the imaging evidence is strong, but the biochemical evidence is weak. Biochemical evidences supporting the protein-protein and protein-mtDNA interactions involved in the axis of this study, the mtDNA-Mic60-Miro complex, are especially desired.

2) Since little biochemical evidence was provided, some statements and the working model may appear overstated. For instance, Miro is known to be an outer membrane protein, how it interacts with the MICOS complex is not well understood. In Fig 6, it seems that Miro crosses the outer membrane, which is not expected based on available literature.

The authors are therefore recommended to make some effort on mito-ER contacting site purification and further validate their findings using biochemical assays. That would significantly strengthen the paper.

3) Besides what is presented in the current manuscript, have the authors examined the effects of mitochondrial fission/fusion genes, such as Mfn1/2, Drp1, and Opa1.

Reviewer #2 (Remarks to the Author):

In this manuscript titled "ER-mitochondria contacts promote mtDNA nucleoids active transportation via mitochondrial dynamic tubulation", Jinshan Qin et al. demonstrate a mechanism for mitochondrial nucleoids transport and distribution. They showed that EMCS and KIF5B-Miro1-MICOS axis determine the initiation of MDT and subsequent mitochondrial nucleoids transport. The findings shown in this manuscript are potentially interesting to the readers. However, the important finding about the role of Miro1 in ER-mitochondria contact and the interaction between Miro1 and MICOS have been published by another lab (Souvik Modi et al, Nat Commun, 2019), some data were preliminary, additional experiments are needed before publication. Therefore, the manuscript could not be accepted now.

Major comments

1. MICOS complex is highly associated with mtDNA nucleoids organization and distribution, but the mechanism is still unclear. The interaction between MICOS and mtDNA is the key bridge for mitochondrial nucleoids transport. Thus, the authors should show strong evidence for interaction between MICOS and mtDNA nucleoids, whether it is direct or indirect interaction. In addition, whether the other MICOS subunit such as Mic19 or Mic10 regulate mitochondrial nucleoids transport?
2. The manuscript showed that MDT often initiates at the EMCS (Fig 2), what is the role of mitochondria-ER contact in MDT? In addition, what role does mitochondria-ER contact play in mitochondrial nucleoids transport? The title of manuscript showed "ER-mitochondria contacts promote mtDNA nucleoids active transportation", but all manuscript just showed that MDT often initiates at the EMCS, additional solid data are needed to further prove this conclusion. The author can deplete some key factors linking ER-mitochondria contact to check MDT activity and mitochondrial nucleoids transport.

Minor comments

1. About perinuclear localization, all the related imagings (Fig 5a, 5b, 5h, and supplementary Fig 6c, 6d) were lack of DAPI staining to mark the nucleus.
2. Supplementary Fig. 8e, Western blot by anti-Miro1 antibody is not clear, the quality of Western blot are needed to be improved.

Reviewer #3 (Remarks to the Author):

Qin et al. describe movements of mtDNA in MDTs, which are tubes emanating from mitochondria. These tubes can be observed in live cells where they project from mitochondria and occasionally connect with other mitochondria, carrying proteins and other small molecules between mitochondria without full-fledged fusion. However, nucleoids were thought to be excluded because of their size. Here the authors show that nucleoids can be transported in MDTs, they show that this transport is KIF5B dependent, as previously also shown for other

types of MDT transport, and they show that transport is affected by MICOS and correlates with Miro proteins, suggesting that these proteins connect the transport of nucleoids to the actions of KIF5B. Furthermore, the authors suggest that many of the MDTs are formed at ER-mito contact sites. Importantly, their results suggest that MDTs can distribute and/or exchange nucleoids between mitochondria without fully fusing the mitochondria.

This manuscript contains a number of interesting ideas that are worth testing. Unfortunately, many of the points made in this manuscript are not rigorously established. The paper also suffers from some confusion as to what actually constitutes an MDT.

General comments:

The MDTs are described here as branches that grow from the tips or sides of mitochondria, but some are very thin and others seem to have the same diameters as normal mitochondria. The authors should categorize the MDTs and determine which functions can be attributed to which type of MDT. Other labs have called the thinner processes “mitochondrial nanotunnels”. These are devoid of cristae, which would preclude the functions of MICOS, and they seem too narrow to allow for transport of nucleoids. The thicker branches are similar in diameter to normal mitochondria and may in fact just be protrusions that grow into a branch of a mitochondria. This phenomenon is often observed with conventional live cell microscopy. Movements of nucleoids in these branches is worth documenting. Observing nucleoids in the thinner tubes (nanotunnels) would be interesting. Because both processes are potentially important for mixing mtDNAs, their roles in nucleoid distribution need to be rigorously tested as separate phenomena and then compared and contrasted with exchange of nucleoids through conventional fission and fusion.

Specific points:

Most of the experiments in this paper are descriptive. In the few cases where the system was manipulated (KIF5B KO and Mic60 siRNA), the answers seem trivial.

Fig. 1. The idea that the tips of protrusions often contain nucleoids that may drive tubule formation is interesting, but this should be tested for example by strengthening or disrupting the MAM. There is no indication whether the thin mitochondrial tubules (nanotunnels) also contain nucleoids.

Fig. 2. The two examples for ER tubules at the sites of MDT appear to show the formation of “nanotunnels”, as described by others. The authors need more data for statistical relevance of the coupling to ER. The authors should also distinguish events that give rise to thin protrusions, as shown here, and protrusions with normal diameters. The authors should also test the relevance of the purported coupling by manipulating the ER contact sites.

Fig. 3. Enrichment of Miro1 and mtDNA at ECMS has been observed by others.

Fig. 4. Effects of mic60 siRNA on transport and ECMS are likely indirect. This protein is important for maintaining mitochondrial morphology through its effects on cristae. Any disruption of this would affect transport and coupling to ER. The title of this figure saying that mic60 links mtDNA to Miro, Kif5B and EMCS is an overstatement.

Fig. 5. Effects of Mic60 siRNA on the peripheral distribution of mitochondria and the nucleoids can be indirect. Transfer of mtDNA between mitochondria is not shown. Do the MDTs with nucleoids at their tips fuse with other mitochondria? This should be documented with statistical relevance.

Minor point: the authors should indicate in the figures what sort of treatments are being tested. Without that, it is difficult to follow.

Author rebuttal, first version:

Reviewer #1 (Remarks to the Author):

Qin and colleagues developed a home-made super-resolution microscope, GI-SIM and applied it into the study of mitochondrial DNA nucleoids transportation. They described that the ER-mitochondria contact site and the MICOS complex are important in this process. Generally speaking, the study was well designed, performed and controlled. The major technology here is very unique and provides clear and convincing images. I will support further consideration of this manuscript for publication in Nature communication.

Author reply:

We thank the reviewer for finding our work of interest and potential. We would also like to thank the reviewer for the constructive and detailed comments that have helped to strengthen and clarify our revised manuscript.

Listed below are a few points that will help improve the manuscript.

1) Overall the imaging evidence is strong, but the biochemical evidence is weak. Biochemical evidences supporting the protein-protein and protein-mtDNA interactions involved in the axis of this study, the mtDNA-Mic60-Miro complex, are especially desired.

Author reply:

We thank the reviewer for the comment on the biochemical evidence for protein-protein and protein-mtDNA interactions, which would definitely reinforce the model we proposed for mtDNA transportation. In response to the reviewer's suggestion, we have conducted co-immunoprecipitation experiments and added the results as Fig. 4a in the revised manuscript.

In order to prove the interaction between Mic60 and mtDNA, we performed co-immunoprecipitation experiments between endogenous Mic60 and TFAM. In mammalian cells, mtDNA is protein-coated and packaged into nucleoids¹ and TFAM protein is a main component of the nucleoid^{2, 3}. As shown in **Fig. R1**, TFAM was specifically co-immunoprecipitated with Mic60 in lysates from HEK293T cells. Mic60 is also reportedly found in purified HeLa mtDNA⁴ and with direct interaction with TFAM⁵. Together, these lines of evidence support that Mic60 interacts with mtDNA.

Fig. R1 Co-immunoprecipitation of Mic60 with endogenous TFAM. Immunoprecipitation was performed from HEK293T cells, and the immunoprecipitates were blotted as indicated. Input: 2%.

Meanwhile, we also confirmed the interaction between endogenous Miro1 and Mic60 using co-immunoprecipitation in HEK293T cells (Fig. R2). Our result is consistent with the published work reporting that Miro1 interacts with Mic60^{6, 7}.

Fig. R2 Co-immunoprecipitation of Mic60 with endogenous Miro1. Immunoprecipitation was performed from HEK293T cells, and the immunoprecipitates were blotted as indicated. Input: 2%.

2) Since little biochemical evidence was provided, some statements and the working model may appear overstated. For instance, Miro is known to be an outer membrane protein, how it interacts with the MICOS complex is not well understood. In Fig 6, it seems that Miro crosses the outer membrane, which is not expected based on available literature. The authors are therefore recommended to make some effort on mito-ER contacting site purification and further validate their findings using biochemical assays. That would significantly strengthen the paper.

Author reply:

We thank the reviewer for the comment and suggestion about the interaction between Miro1 and the MICOS complex. Besides the colocalization analysis in the original manuscript, in this revised manuscript, we have performed biochemical assays to confirm

the interaction between Miro1 and Mic60, a subunit of the MICOS complex, using co-immunoprecipitation of endogenous Miro1 and Mic60 in HEK293T cells (Fig. R2). Regarding the possible mechanism for the interaction between Miro1 and Mic60, Miro1 was reported to contain a C-terminal transmembrane domain to target mitochondria (Fig.

R3)⁸. This transmembrane domain may extend into the intermembrane space of mitochondria. Mic60 was also reported to contain a transmembrane domain that spans the inner mitochondrial membrane with the bulk of the protein protruding into the intermembrane space (Fig. R4)^{9, 10}. Therefore, these lines of evidence provide a possible mechanism that the transmembrane domains of Miro1 and Mic60 mediate the interaction between Miro1 and the MICOS complex in the mitochondrial intermembrane space.

[Redacted]

Fig. R3 Image redacted. Schematic illustration of the primary structure of Miro. See reference 8

[Redacted]

Fig. R4 Image redacted: Fig. R4 was taken from Figure 1b in reference 10.

Regarding the suggestion that “to make some effort on mito-ER contacting site purification and further validate their findings using biochemical assays”, in the original manuscript, we have shown that knockdown of Mic60 (Fig. 4d-h) could destabilize EMCS and affect MDT-based nucleoids transport. Loss of Mic60 reduces Miro1 at the EMCS (Supplementary Fig. 9e) and Miro1 can interact with Mfn2¹¹, which is a potential tether between ER and mitochondria in mammalian cells^{12, 13}. Despite these findings, the molecular nature of the physical tether between ER and mitochondria is actually unclear

in mammalian cells¹⁴. The formal and univocal demonstration that the lack of a given molecule abolishes ER-mitochondria contacts has not been provided¹⁴. Most likely, different and independent tethering complexes may exist and compensate one for the lack of the others, increasing the complexity of the analysis¹⁴. For example, when the ER-mitochondrial tether Mfn2 was depleted, mitochondria still maintained ER contact over

time and no significant changes were detected in the ability of division factor Mff punctae to localize to positions of ER-mitochondrial contacts or ER-mediated mitochondrial constriction¹⁵. Furthermore, the high density of ER and mitochondrial networks in mammalian cells suggests that ER-mitochondria contact sites may greatly outnumber the ER-associated MDT events. Supporting this argument is the observation that Miro1, the key molecule that drives ER-associated MDT, is only concentrated at a small fraction of EMCS (Fig. R5). In line with this observation, a recent study also reported that a really small fraction of ER-mitochondria contact sites are linked to ER-associated division (Fig. R6)¹⁶. Thus, only a small subset of the numerous ER-mitochondria contacts seem to participate in a particular function. This redundancy makes it difficult to validate the molecules specific for MDT using purified mito-ER contact sites.

Fig. R5 The percentage of EMCS concentrated with Miro1. (a) Left, example merged image of a mitochondrion labeled with Mito-DsRed (green), ER labeled with mEmerald-sec61 β (magenta) and Miro1 labeled with α -Miro1 antibody in a fixed Cos-7 cell. Right, thresholded image from (a). White arrows indicate colocalization regions and orange arrows indicate Miro1. Scale bar: 2 μ m. (b) The percentage of EMCS that colocalize with Miro1 and without Miro1, n=775.

[Redacted]

Image redacted: Fig. R6 The percentage of persistent mitochondrial-ER colocalizations that become sites of mitochondrial constriction or division over 5 min in live U2OS cells. See reference 16.

4

3) Besides what is presented in the current manuscript, have the authors examined the effects of mitochondrial fission/fusion genes, such as Mfn1/2, Drp1, and Opa1.

Author reply:

We thank the reviewer for the great suggestion regarding mitochondrial fission/fusion proteins. In this revised manuscript, we have examined the effects of mitochondrial fusion and fission on MDT-dependent mitochondrial nucleoids transport and added the new data as Supplementary Fig. 4.

Briefly, to test whether MDT-based mitochondrial nucleoids transport is independent on mitochondrial fission, we depleted Drp1 in MEF cells using siRNA and co-transfected these cells with TOM20-GFP and TFAM-mCherry. Drp1 was substantially depleted in Drp1 RNAi cells in comparison with the control cells (Fig. R7b). In Drp1 RNAi cells, we found that a small number of mitochondrial nucleoids were present in the form of enlarged clusters within aberrant and elongated mitochondria, a morphological feature normally observed when Drp1 is depleted (Fig. R7a). Nevertheless, in these Drp1-depleted cells, we were still able to observe frequent MDT events which could actively transport the nucleoids (Fig. R7c). This observation is in line with a previous study reporting that the aggregation and mislocalization of nucleoids caused by depletion of the MICOS complex could not be fully rescued by Drp1 over-expression¹⁷. Similarly, we could also observe frequent events of MDT-mediated nucleoids transportation in MEF cells down-regulated with Mfn1 and Mfn2 (Fig. R7f), despite that the mitochondria network was completely disrupted and turned to be round or oval-shaped vesicles with a large fraction of mitochondria devoid of nucleoids (Fig. R7d, e). This observation is in agreement with our previous work (Fig. R8)¹⁸, which showed that dynamic tubulation can occur in Mfn-null cells.

Collectively, MDT-mediated nucleoids transport is a new mechanism, which is independent on mitochondrial fission and fusion. Nonetheless, these three mitochondrial processes may coordinate to properly distribute nucleoids in the mitochondrial network. As segregation of duplicated mtDNA is coupled with mitochondrial division¹⁶, blocking mitochondrial division can result in abnormally enlarged nucleoids, which are possibly of hampered transportation in the thin MDT tubules. When mitochondrial fusion is blocked in the cell, even though MDT can still normally occur, the nucleoids transported by MDT would not be possible to exchange among different mitochondria without mitochondrial fusion, thus leading to a large fraction of mitochondria devoid of nucleoids (Fig. R7d).

Therefore, mitochondrial fission, dynamic tubulation, and fusion may act in turn to facilitate the proper distribution in the mitochondrial network.

Fig. R7 MDT-mediated nucleoids transport is independent on mitochondrial fission and fusion. (a) Control and Drp1 siRNA MEF cells expressing TOM20-GFP and TFAM-mCherry. Scale bar: 10 μ m. (b) Western blots of Drp1 indicate depletion of Drp1 in lysates of cells transfected with siRNA against Drp1. (c) Time-lapse sequence of MDT-mediated nucleoids transport in Drp1 knockdown MEF cells. White arrows mark the tips of tubules generated by the MDT processes. Orange arrows indicate the nucleoids. Scale bar, 2 μ m. (d) Wide-type (WT) and Mfn-null MEF cells expressing TOM20-GFP and TFAM-mCherry. Scale bar, 10 μ m. (e) Mfn1 and Mfn2 levels in Mfn-null and WT MEF cells as shown by western blotting. (f) Time-lapse sequence of MDT-mediated nucleoids transport in Mfn-null MEF cells. White arrows mark the tips of tubules generated by the MDT processes. Orange arrows indicate nucleoids. Scale bar, 2 μ m.

Fig. R8 Time-lapse sequence of mitochondrial dynamic tubulation in WT (top row) and Mfn-null MEF cells (bottom row). Scale bar, 1 μ m.

Reviewer #2 (Remarks to the Author):

In this manuscript titled “ER-mitochondria contacts promote mtDNA nucleoids active transportation via mitochondrial dynamic tubulation”, Jinshan Qin et al. demonstrate a mechanism for mitochondrial nucleoids transport and distribution. They showed that EMCS and KIF5B-Miro1-MICOS axis determine the initiation of MDT and subsequent mitochondrial nucleoids transport. The findings shown in this manuscript are potentially interesting to the readers. However, the important finding about the role of Miro1 in ER-mitochondria contact and the interaction between Miro1 and MICOS have been published by another lab (Souvik Modi et al, Nat Commun, 2019), some data were preliminary, additional experiments are needed before publication. Therefore, the manuscript could not be accepted now.

Author reply:

We thank the reviewer for finding our work of interest and potential. We would also like to thank the reviewer for the constructive and detailed comments that have helped to strengthen and clarify our revised manuscript.

Major comments

1. MICOS complex is highly associated with mtDNA nucleoids organization and distribution, but the mechanism is still unclear. The interaction between MICOS and mtDNA is the key bridge for mitochondrial nucleoids transport. Thus, the authors should show strong evidence for interaction between MICOS and mtDNA nucleoids, whether it is direct or indirect interaction. In addition, whether the other MICOS subunit such as Mic19 or Mic10 regulate mitochondrial nucleoids transport?

Author reply:

We thank the reviewer for the comment on the interaction between MICOS and mtDNA. In addition to the immunofluorescence imaging data showing that mtDNA and Mic60, the core subunit of the MICOS complex, are spatially colocalized (Fig. 4b and Supplementary Fig. 9a), we have conducted co-immunoprecipitation experiments in this revised manuscript to provide biochemical evidence for their interactions (Fig. R1), which reinforces the model we proposed for mtDNA transportation.

In order to prove the interaction between Mic60 and mtDNA, we performed co-immunoprecipitation experiments between endogenous Mic60 and TFAM. In

mammalian, mtDNA is protein-coated and packaged into nucleoids¹ and TFAM protein is a main component of the nucleoid^{2, 3}. As shown in **Fig. R1**, TFAM was specifically co-immunoprecipitated with Mic60 in lysates from HEK293T cells. Mic60 is also reportedly found in purified HeLa mtDNA⁴ and with direct interaction with TFAM⁵. Together, these lines of evidence support that MICOS interacts with mtDNA.

Fig. R1 Co-immunoprecipitation of Mic60 with endogenous TFAM. Immunoprecipitation was performed from HEK293T cells, and the immunoprecipitates were blotted as indicated. Input: 2%.

Meanwhile, we also confirmed the interaction between endogenous Miro1 and Mic60 using co-immunoprecipitation in HEK293T cells (Fig. R2). Our result is consistent with the published work reporting that Miro1 interacts with Mic60^{6,7}.

Fig. R2 Co-immunoprecipitation of Mic60 with endogenous Miro1. Immunoprecipitation was performed from HEK293T cells, and the immunoprecipitates were blotted as indicated. Input: 2%.

Regarding whether the other MICOS subunit such as Mic19 or Mic10 regulates mitochondrial nucleoids transport, we have sought for answers in the literatures. Mic19 is one of the core components of the MICOS complex, and downregulation of Mic19 results in instability of other MICOS components and disassembly of the MICOS complex¹⁷. While the interaction between Mic19 and mitochondrial nucleoids has not been confirmed, Mic19 has been reported to interact directly with Mic60 and knockdown of Mic19 results in remarkable downregulation of Mic60 (Fig. R3b)¹⁷. Besides, knockdown of Mic19 also induces enlarged mitochondrial nucleoids possibly due to the downregulation of Mic60¹⁷. Importantly, Mic19 contains a transmembrane domain that spans the inner mitochondrial membrane and protrudes into the intermembrane space (Fig. R4)¹⁰, and immunoprecipitation in HeLa cells has revealed robust interactions between Mic19 and Miro1⁶. Therefore, Mic19 is likely to play a part in regulating mitochondrial nucleoids transport. In comparison with Mic19 and Mic60, Mic10 is a small integral intermembrane protein¹⁹ without protrusion in the

intermembrane space (Fig. R4)¹⁰, suggesting that Mic10 is unlikely to interact with Miro1 on the outer membrane. Moreover, while downregulation of Mic60 greatly decreases the protein level of Mic10 (Fig. R3a), knockdown of Mic10 does not affect the level of Mic60 (Fig. R3c). Importantly, Mic10 knockdown has no effect on the size and distribution of the nucleoids¹⁷. These data suggest that Mic10 has minimal

effect on mitochondrial nucleoids transport through the mechanism we proposed.

Fig. R3 Western blot analysis of Mic60/Mitofilin, Mic19/CHCHD3, Mic10/MINOS1 and Mic25/CHCHD6 in control, shMic60 (a), shMic19 (b) or shMic10 (c) MEFs. TOM20 and GAPDH served as protein-loading control¹⁷.

[redacted]

Fig. R4 Image redacted: Fig. R4 was taken from Figure 1b in reference 10.

2. The manuscript showed that MDT often initiates at the EMCS (Fig 2), what is the role of mitochondria-ER contact in MDT? In addition, what role does mitochondria-ER contact play in mitochondrial nucleoids transport? The title of manuscript showed “ER-mitochondria contacts promote mtDNA nucleoids active transportation”, but all manuscript just showed that MDT often initiates at the EMCS, additional solid data are needed to further prove this conclusion. The author can deplete some key factors linking ER-mitochondria contact to check MDT activity and mitochondrial nucleoids transport.

Author reply:

We thank the reviewer for the questions on the roles of EMCS in MDT and mitochondrial nucleoids transport. As for its role in promoting MDT, increasing evidence suggests that interorganellar membrane contacts could form membrane microdomains with specialized lipid and protein components²⁰. In this capacity, ER-mitochondria contacts could facilitate to create a spatial platform on mitochondria that selectively recruits the protein required for MDT, such as Miro1 (Supplementary Fig. 7a). Here, we found that Miro1,

the KIF5B receptor on mitochondria, is enriched at the EMCS (Fig. 3c). The enrichment of Miro1 recruits more KIF5B, the motor protein we previously proved to drive MDT¹⁸, and makes EMCS a hot spot for MDT. Regarding the molecular mechanism, it would be an interesting research subject to dissect the details of lipid and protein components in the membrane microdomains at EMCS in the future.

As for the role of EMCS in mitochondrial nucleoids transport, in addition to promoting MDT by enriching Miro1 and KIF5B, EMCS is also known to coordinate mtDNA replication and mark the division sites where nascent mtDNA segregates to daughter mitochondria¹⁶. In this work, we found that Miro1 foci and nucleoids are frequently co-localized at the EMCS (Fig. 3d). Therefore, EMCS creates a platform for establishing the KIF5B-Miro1-MICOS-nucleoids axis to promote mitochondrial nucleoids transport.

Regarding the suggestion that “deplete some key factors linking ER-mitochondria contact to check MDT activity and mitochondrial nucleoids transport”, in the original manuscript, we have performed knockdown of Miro1. It was also reported that the homologous protein of Miro1 in yeast, Gem1, is localized at EMCS and regulates ER-mitochondria contacts²¹. Recently, Miro was also reported to regulate the number of EMCS in mammalian cell⁶. Miro1 can interact with Mfn2¹¹, which is a potential tether between ER and mitochondria in mammalian cells^{12, 13}. Upon Miro1 knockdown, we observed that the MDT activity was greatly repressed in the cell (Supplementary Fig. 7a). Interestingly, we also found that Mic60 knockdown could decrease the stability of EMCS (Fig. 4d-h), which might be because that loss of Mic60 reduces Miro1 at the EMCS (Supplementary Fig. 9e).

Despite these findings, the molecular nature of the physical tether between ER and mitochondria is actually unclear in mammalian cells¹⁴. The formal and univocal demonstration that the lack of a given molecule abolishes ER-mitochondria contacts has not been provided¹⁴. Most likely, different and independent tethering complexes may exist and compensate one for the lack of the others, increasing the complexity of the analysis¹⁴. For example, when the ER-mitochondrial tether Mfn2 was depleted, mitochondria still maintained ER contact over time and no significant changes were detected in the ability of division factor Mff punctae to localize to positions of ER-mitochondrial contacts or ER-mediated mitochondrial constriction¹⁵. Furthermore, the high density of ER and mitochondrial networks in mammalian cells suggests that ER-mitochondria contact sites

may greatly outnumber the ER-associated MDT events. Supporting this argument is the observation that Miro1, the key molecule that drives ER-associated MDT, is only concentrated at a small fraction of EMCS (Fig. R5). In line with this observation, a recent study also reported that a really small fraction of ER-mitochondria contact sites are linked to ER-associated division (Fig. R6)¹⁶. Thus, only a small subset of the numerous ER-mitochondria contacts seem to participate in a particular function. This redundancy makes

it difficult to validate the molecules specific for MDT and mitochondrial nucleoids transport by globally disrupting mito-ER contact sites.

Fig. R5 The percentage of EMCS concentrated with Miro1. (a) Left, example merged image of a mitochondrion labeled with Mito-DsRed (green), ER labeled with mEmerald-sec61 β (magenta) and Miro1 labeled with α -Miro1 antibody in a fixed Cos-7 cell. Right, thresholded image from (a). White arrows indicate colocalization regions and orange arrows indicate Miro1. Scale bar: 2 μ m. (b) The percentage of EMCS that colocalize with Miro1 and without Miro1, n=775.

[Redacted]

Image redacted. Fig. R6 The percentage of persistent mitochondrial-ER colocalizations that become sites of mitochondrial constriction or division over 5 min in live U2OS cells. See reference 16.

Minor comments

1. About perinuclear localization, all the related images (Fig 5a, 5b, 5h, and supplementary Fig 6c, 6d) were lack of DAPI staining to mark the nucleus.

Author reply:

We thank the reviewer for raising the concern. During the experiments, we collected bright field images to outline the nuclei as shown in Supplementary Fig 7c, 7d. We found that in KIF5B KO or Miro KD cells, mitochondria network retracted and surrounded the nuclei.

2. Supplementary Fig. 8e, Western blot by anti-Miro1 antibody is not clear, the quality of Western blot are needed to be improved.

Author reply:

We thank the reviewer for raising the concern. The Western blot quality was due to the unideal performance of the anti-Miro1 antibody. To compensate the smearing effect, we used ImageJ to integrate the Western blot band intensity which made the conclusion that Mic60 knockdown reduces Miro1 more quantitatively. With β -actin normalization, the analysis showed that the reduction of Miro expression upon Mic60 knockdown was 41% in comparison with the control cells. We have added the analysis into **Supplementary Fig 9e** in the revised manuscript.

Reviewer #3 (Remarks to the Author):

Qin et al. describe movements of mtDNA in MDTs, which are tubes emanating from mitochondria. These tubes can be observed in live cells where they project from mitochondria and occasionally connect with other mitochondria, carrying proteins and other small molecules between mitochondria without full-fledged fusion. However, nucleoids were thought to be excluded because of their size. Here the authors show that nucleoids can be transported in MDTs, they show that this transport is KIF5B dependent, as previously also shown for other types of MDT transport, and they show that transport is affected by MICOS and correlates with Miro proteins, suggesting that these proteins connect the transport of nucleoids to the actions of KIF5B. Furthermore, the authors suggest that many of the MDTs are formed at ER-mito contact sites. Importantly, their results suggest that MDTs can distribute and/or exchange nucleoids between mitochondria without fully fusing the mitochondria.

This manuscript contains a number of interesting ideas that are worth testing. Unfortunately, many of the points made in this manuscript are not rigorously established. The paper also suffers from some confusion as to what actually constitutes an MDT.

Author reply:

We thank the reviewer for finding our work of interest and potential. We would also like to thank the reviewer for the constructive and detailed comments that have helped to strengthen and clarify our revised manuscript.

General comments:

The MDTs are described here as branches that grow from the tips or sides of mitochondria, but some are very thin and others seem to have the same diameters as normal mitochondria. The authors should categorize the MDTs and determine which functions can be attributed to which type of MDT. Other labs have called the thinner processes “mitochondrial nanotunnels”. These are devoid of cristae, which would preclude the functions of MICOS, and they seem too narrow to allow for transport of nucleoids. The thicker branches are similar in diameter to normal mitochondria and may in fact just be protrusions that grow into a branch of a mitochondria. This phenomenon is often observed with conventional live cell microscopy. Movements of nucleoids in these branches is worth documenting. Observing nucleoids in the thinner tubes (nanotunnels) would be interesting. Because both processes are potentially important for mixing mtDNAs, their roles in nucleoid distribution need to be rigorously tested as separate phenomena and then compared and contrasted with exchange of nucleoids through conventional fission and fusion.

Author reply:

We thank the reviewer for the comments and suggestions on the mitochondrial thinner tubes (nanotunnels) as well as conventional fission and fusion of mitochondria. We will address them separately in the following.

In our previous work, we defined MDT as the dynamic tubulation process driven by KIF5B along microtubules¹⁸, and thus we focused more on the tubular elongation behavior rather than the tubular diameter. Nevertheless, we totally agree with the reviewer that “The thinner tubes or nanotunnels may be too narrow to allow for transport of nucleoids”. Following the reviewer’s suggestion, we re-analyzed the data and documented the diameter of the tubules generated by MDT. As shown in **Figure R1a**, GI-SIM fluorescence imaging indicates that MDT tubules were ranged from 80 to 350 nm in diameter (the lower bound is likely limited by the spatial resolution of GI-SIM). Based on a previous work reporting that mitochondrial nanotunnels vary between 40 and 200 nm in diameter²², about 61% (36/59) of the tubules in **Figure R1a** can fall in the category of nanotunnel. Among all MDT-generated tubules, we next pooled together those that contained nucleoids (**Fig. R1b**) and noticed two interesting points. Firstly, only 22% (8/36) of the tubules ranging from 40 to 200 nm in diameter contained nucleoids, while 70% (16/23) of those thicker than 200 nm contained nucleoids. Secondly, the minimal diameter of MDT with nucleoids is 140 nm, larger than that of nucleoids, which was measured by STED imaging as a defined, uniform mean size of ~100 nm in mammals³. Interestingly, we also noticed that the tubule size could change dynamically during MDT with the presence of nucleoids. As exemplified in **Fig 1f**, MDT pulled out a thin tubule without a nucleoid at the beginning. Then along with the elongation of the tubule, the tubular diameter increased gradually, and the nucleoid was transported in the tubule. Another example is given in **supplementary Fig 2b**. We observed that the nucleoid was paused at the junction between the thick and thin parts of the tubule and could not run into the thin part until it became thicker.

These analyses indicate that nucleoids indeed have lower chances to be transported in thinner tubules. We have added these analyses in the revised manuscript.

Fig. R1 The diameter distribution of total MDT and MDT with nucleoids. (a) The diameter distribution of total MDT, $n=59$ events from 10 cells. (b) The diameter distribution of MDT with the nucleoids, $n=24$ events from 10 cells. (c) The percentage of nucleoids-containing tubules for different MDT tubular diameters. $n=36$ for MDT with diameter ≤ 200 nm, $n=23$ for MDT with diameter > 200 nm.

Regarding the difference between MDT and mitochondrial fission/fusion, MDT is a new mechanism for mtDNA partition and transportation that is distinct from mitochondrial fusion or fission. Firstly, in this revised manuscript, we have examined the effects of mitochondrial fusion and fission on MDT-dependent mitochondrial nucleoids transport (Fig. R2) and added the new data as Supplementary Fig. 4. The data showed that MDT-mediated nucleoids transport is independent on mitochondrial fission and fusion. MDT events could be frequently observed to transport nucleoids in Drp1 or Mfn depleted cells (Fig. R2). Secondly, the MDT activity can explain phenomena in mitochondrial dynamics such as side branching during the establishment of peripheral mitochondrial network, which are hard to interpreted based on the traditional mitochondrial fusion/fission activities. Inhibiting MDT-mediated active nucleoids transportation can result in significant loss of nucleoids within the mitochondrial network in the peripheral zone of the cell (Fig. 5j), an area so wide that cells can benefit from active transportation of mtDNA. Thirdly, MDT-based mtDNA partition and transportation could be considered as ‘reaching out for help’, a concept proposed in a

previous work²². Thus, the active mtDNA transportation model is especially crucial for cells with restricted mitochondrial motility, such as skeletal and

cardiac muscle cells.

Fig. R2 MDT-mediated nucleoids transport is independent on mitochondrial fission and fusion. (a) Control and Drp1 siRNA MEF cells expressing TOM20-GFP and TFAM-mCherry. Scale bar: 10 μ m. (b) Western blots of Drp1 indicate depletion of Drp1 in lysates of cells transfected with siRNA against Drp1. (c) Time-lapse sequence of MDT-mediated nucleoids transport in Drp1 knockdown MEF cells. White arrows mark the tips of tubules generated by the MDT processes. Orange arrows indicate the nucleoids. Scale bar, 2 μ m. (d) Wide-type (WT) and Mfn-null MEF cells expressing TOM20-GFP and TFAM-mCherry. Scale bar, 10 μ m. (e) Mfn1 and Mfn2 levels in Mfn-null and WT MEF cells as shown by western blotting. (f) Time-lapse sequence of MDT-mediated nucleoids transport in Mfn-null MEF cells. White arrows mark the tips of tubules generated by the MDT processes. Orange arrows indicate nucleoids. Scale bar, 2 μ m.

In the meantime, while the MDT-mediated transport of mitochondrial nucleoids is independent on mitochondrial fission and fusion, these three mitochondrial processes may coordinate to properly distribute nucleoids in the mitochondrial network. As segregation of duplicated mtDNA is coupled with mitochondrial division¹⁶. When blocking

mitochondrial division, the abnormally enlarged nucleoids are possibly of hampered transportation in the thin MDT tubules. Additionally, we found that a subset of division events occurred at the initiation sites of MDT when the MDT tubule was extending from the mother mitochondrion (Fig R3). This suggests that the MDT process could facilitate the partition of nucleoids into the daughter mitochondrion before fission. When mitochondrial fusion is blocked in the cell, even though MDT can still normally occur, the nucleoids transported by MDT would not be possible to exchange among different mitochondria without mitochondrial fusion, thus leading to a large fraction of mitochondria devoid of nucleoids (Fig. R2d). Therefore, mitochondrial fission, dynamic tubulation, and fusion may act in turn to facilitate the proper distribution in the mitochondrial network.

Fig. R3 Mitochondrial fission can occur at the initiation sites of MDT when the tubule is extending. Two time-lapse images in cells expressing mito-YFP. The red sparks indicate the sites of fission. Arrows indicate thin tubules pulled out of the mitochondria. Scale bar: 5 μ m

Specific points:

Most of the experiments in this paper are descriptive. In the few cases where the system was manipulated (KIF5B KO and Mic60 siRNA), the answers seem trivial.

Fig. 1. The idea that the tips of protrusions often contain nucleoids that may drive tubule formation is interesting, but this should be tested for example by strengthening or disrupting the MAM. There is no indication whether the thin mitochondrial tubules (nanotunnels) also contain nucleoids.

Author reply:

We thank the reviewer for the comments. That “the tips of protrusions often contain nucleoids that may drive tubule formation” is mainly because ER-mitochondria contact sites (EMCS) function as a platform for establishing the KIF5B-Miro1-MICOS-nucleoids axis to promote mitochondrial nucleoids transport. EMCS is known to spatially linked to

nucleoids in general¹⁶. Meanwhile, EMCS can recruit and enrich Miro1 (Fig. 3c), which recruits more KIF5B, the motor protein we previously proved to drive MDT¹⁸, and makes EMCS a hot spot for MDT. As a result, Miro1 foci and nucleoids are frequently co-localized at EMCS (Fig. 3d).

As for the molecular mechanism, EMCS is likely equivalent to mitochondria-associated membrane (MAM), a subdomain of the endoplasmic reticulum apposed to mitochondria that comprises a unique set of proteins interacting with mitochondrial proteins²³. It would be an interesting research subject to dissect the details of lipid and protein components in the membrane microdomains at EMCS in the future. In this work, we showed that knockdown of Miro1 decreases the MDT frequency as well as MDT-based nucleoids transportation (Supplementary Fig. 7a). A previous study has proven that depletion of Miro1 decreases the stability of EMCS⁶. Miro1 can interact with Mfn2¹¹, which is a potential tether between ER and mitochondria in mammalian cells^{12, 13}. Interestingly, we also found that Mic60 knockdown could decrease the stability of EMCS and abolish the active transportation of nucleoids (Fig. 4d-h and Fig. 5e-f), which might be because that loss of Mic60 reduces Miro1 at the EMCS (Supplementary Fig. 9e). Together, these data suggest that knockdown of Miro1 or Mic60 can destabilize MAM and affect MDT-based nucleoids transportation.

To check whether the thin mitochondrial tubules (nanotunnels) also contain nucleoids, we have re-analyzed the data and documented the diameter of the tubules generated by MDT in this revised manuscript. We found that only 22% (8/36) of the tubules ranging from 40 to 200 nm in diameter (the range measured for mitochondrial nanotunnels²²) contained nucleoids, while 70% (16/23) of those thicker than 200 nm contained nucleoids. We also noticed that none of the MDT tubules thinner than 140 nm contained nucleoids (Fig R1). These analyses indicate that nucleoids indeed have lower chances to be transported in thinner tubules.

Fig. 2. The two examples for ER tubules at the sites of MDT appear to show the formation of “nanotunnels”, as described by others. The authors need more data for statistical relevance of the coupling to ER. The authors should also distinguish events that give rise to thin protrusions, as shown here, and protrusions with normal diameters. The authors should also test the relevance of the purported coupling by manipulating the ER contact sites.

Author reply:

We thank the reviewer for the suggestions. We observed that 90% MDT events occurred at the sites of contact between ER and mitochondria (n = 51 from 23 cells) (Fig. 2a, b; Supplementary Fig. 5). Following the reviewer's suggestion, we have tried to

distinguish events that give rise to thin protrusions (nanotunnels) and thick protrusions (Fig. R4). We re-analyzed the data and found that 92% (24/26) of the tubules ranging from 40 to 200 nm in diameter (mitochondrial nanotunnels) initiated near EMCS, and 88% (22/25) of those thicker than 200 nm occurred at the sites of ER-mitochondrial contacts (Supplementary Fig. 5c, 6c). There is no significant difference between thin protrusions and thick protrusions initiating at the EMCS. We have added these analyses in the revised manuscript.

MDT categorization	ER tubules mark the initiation sites of MDT
MDT diameter \leq 200nm	92% (24/26)
MDT diameter $>$ 200 nm	88% (22/25)

Fig. R4 Percentage of MDT with different tubular diameters in live Cos-7 cells that occurred at the EMCS. n=26 for MDT with diameter \leq 200nm, n=25 for MDT with diameter $>$ 200nm.

This high level of coincidence of MDT initiation at EMCS is based on the finding that EMCS can recruit and enrich Miro1 (Fig. 3c), which recruits more KIF5B, the motor protein we previously proved to drive MDT¹⁸, and makes EMCS a hot spot for MDT. Furthermore, the coupling to ER is also supported by the observation that knockdown of Miro1 decreases the MDT frequency (Supplementary Fig. 7a). A previous study has proven that depletion of Miro1 decreases the stability of EMCS⁶. Miro1 can interact with Mfn2¹¹, which is a potential tether between ER and mitochondria in mammalian cells^{12, 13}. Together, these data suggest that MDT is coupled to ER at the ER-mitochondria contact sites.

To test the relevance of the purported coupling by manipulating the ER contact sites, we have shown that knockdown of Mic60 could destabilize EMCS and affect MDT-based nucleoids transport. Despite these findings, the molecular nature of the physical tether between ER and mitochondria is actually unclear in mammalian cells¹⁴. The formal and univocal demonstration that the lack of a given molecule abolishes ER-mitochondria contacts has not been provided¹⁴. Most likely, different and independent tethering complexes may exist and compensate one for the lack of the others, increasing the complexity of the analysis¹⁴. For example, when the ER-mitochondrial tether Mfn2 was

depleted, mitochondria still maintained ER contact over time and no significant changes were detected in the ability of division factor Mff punctae to localize to positions of ER-mitochondrial contacts or ER-mediated mitochondrial constriction¹⁵. Furthermore, the high density of ER and mitochondrial networks in mammalian cells suggests that ER-mitochondria contact sites may greatly outnumber the ER-associated MDT events.

Supporting this argument is the observation that Miro1, the key molecule that drives ER-associated MDT, is only concentrated at a small fraction of EMCS (Fig. R5). In line with this observation, a recent study also reported that a really small fraction of ER-mitochondria contact sites are linked to ER-associated division (Fig. R6)¹⁶. Thus, only a small subset of the numerous ER-mitochondria contacts seem to participate in a particular function. This redundancy makes it difficult to validate the molecules specific for MDT and mitochondrial nucleoids transport by globally disrupting mito-ER contacting sites.

Fig. R5 The percentage of EMCS concentrated with Miro1. (a) Left, example merged image of a mitochondrion labeled with Mito-DsRed (green), ER labeled with mEmerald-sec61β (magenta) and Miro1 labeled with α -Miro1 antibody in a fixed Cos-7 cell. Right, thresholded image from (a). White arrows indicate colocalization regions and orange arrows indicate Miro1. Scale bar: 2 μ m. (b) The percentage of EMCS that colocalize with Miro1 and without Miro1, n=775.

[Redacted]

Image redacted. Fig. R6 The percentage of persistent mitochondrial-ER colocalizations that become sites of mitochondrial constriction or division over 5 min in live U2OS cells. See reference 16.

Fig. 3. Enrichment of Miro1 and mtDNA at ECMS has been observed by others. Fig. 4. Effects of mic60 siRNA on transport and ECMS are likely indirect. This protein is important for maintaining mitochondrial morphology through its effects on cristae. Any disruption of this would affect transport and coupling to ER. The title of this figure saying that mic60 links mtDNA to Miro, Kif5B and EMCS is an overstatement.

Author reply:

We thank the reviewer for the comments. In the origin manuscript, we have performed immunofluorescence imaging to show that Mic60 is spatially colocalized with both mtDNA (Fig. 4b and Supplementary Fig. 9a) and Miro1 (Fig. 4c and Supplementary Fig. 9b). We also showed that Mic60 knockdown significantly decreased the endogenous level of Miro1 (Supplementary Fig. 9e). In this revised manuscript, we have performed co-immunoprecipitation experiments to provide biochemical evidence for their interactions (Fig. R7,8), which reinforces the model we proposed for mtDNA transportation.

In order to prove the interaction between Mic60 and mtDNA, we performed co-immunoprecipitation experiments between endogenous Mic60 and TFAM. In mammalian, mtDNA is protein-coated and packaged into nucleoids¹ and TFAM protein is a main component of the nucleoid^{2, 3}. As shown in Fig. R7, TFAM was specifically co-immunoprecipitated with Mic60 in lysates from HEK293T cells. Mic60 is also reportedly found in purified HeLa mtDNA⁴ and with direct interaction with TFAM⁵. Together, these lines of evidence support that MICOS interacts with mtDNA.

Fig. R7 Co-immunoprecipitation of Mic60 with endogenous TFAM. Immunoprecipitation was performed from HEK293T cells, and the immunoprecipitates were blotted as indicated. Input: 2%.

Meanwhile, we also confirmed the interaction between endogenous Miro1 and Mic60 using co-immunoprecipitation in HEK293T cells (Fig. R8). Our result is consistent with the published work reporting that Miro1 interacts with Mic60^{6, 7}.

Fig. R8 Co-immunoprecipitation of Mic60 with endogenous Miro1. Immunoprecipitation was performed from HEK293T cells, and the immunoprecipitates were blotted as indicated. Input: 2%.

Regarding the possible mechanism for the interaction between Miro1 and Mic60, Miro1 was reported to contain a C-terminal transmembrane domain to target mitochondria (Fig. R9)⁸. This transmembrane domain may extend into the intermembrane space of mitochondria. Mic60 was also reported to contain a transmembrane domain that spans the inner mitochondrial membrane with the bulk of the protein protruding into the intermembrane space (Fig. R10)⁹. Therefore, these lines of evidence provide a possible mechanism that the transmembrane domains of Miro1 and Mic60 mediate the interaction between Miro1 and the MICOS complex in the mitochondrial intermembrane space.

[Redacted]

Fig. R9 Image redacted, Fig. R9 Schematic illustration of the primary structure of Miro. See reference 8

[Redacted]

Image redacted: Fig. R10 was taken from Figure 1b in reference 10.

Regarding whether effects of mic60 siRNA on transport and ECMS are indirectly through its effects on cristae, we have sought for the answer from the literatures. Depletion of Mic60 affects MDT-based nucleoids transport and also results in loss of cristae. Actually, knocking down the subunits of the MICOS complex generally leads to disorganized cristae structure and abnormal mitochondrial morphology¹⁰. Mic10 is another main subunit of MICOS complex. In comparison with Mic60, Mic10 is a small integral intermembrane

protein¹⁹ without protrusion in the intermembrane space (Fig. R10)¹⁰, suggesting that Mic10 is unlikely to interact with Miro1 on the outer membrane. Importantly, although knockdown of Mic10 results in reduced cristae and loss of cristae junctions, it has no effects on the size and distribution of the nucleoids¹⁷. This result suggests that disruption of cristae

is not necessarily related with the MDT-based nucleoids transport. Moreover, while downregulation of Mic60 greatly decreases the protein level of Mic10 (Fig. R11a), knockdown of Mic10 does not affect the level of Mic60 (Fig. R11c), implicating the effects of mic60 siRNA on cristae might be indirectly due to loss of Mic10. Together, these data indicate that the effects of mic60 siRNA on nucleoids transport is not merely dependent on its effects on cristae.

Fig. R11 Western blot analysis of Mic60/Mitofilin, Mic19/CHCHD3, Mic10/MINOS1 and Mic25/CHCHD6 in control, shMic60 (a), shMic19 (b) or shMic10 (c) MEFs. TOM20 and GAPDH served as protein-loading control¹⁷.

Fig. 5. Effects of Mic60 siRNA on the peripheral distribution of mitochondria and the nucleoids can be indirect. Transfer of mtDNA between mitochondria is not shown. Do the MDTs with nucleoids at their tips fuse with other mitochondria? This should be documented with statistical relevance.

Author reply:

We thank the reviewer for suggestions. In this study, we unveiled an active partition and transportation mechanism of mitochondrial nucleoids *via* MDT based on the KIF5B-Miro1-MICOS-nucleoids axis. In our previous work, we have shown that MDT is driven by KIF5B motility along microtubules and that fusion of dynamic mitochondrial tubules leads to mitochondrial network formation in the peripheral zone of the cell¹⁸. These features make MDT a particularly efficient way to regulate nucleoids allocation in the mitochondrial network. Here, effects of mic60 siRNA on the peripheral distribution of

mitochondria and the nucleoids are resulted from the disruption of the KIF5B-Miro1-MICOS-nucleoids axis.

Mitochondrial fusion is essential for inter-mitochondria transfer of nucleoids transported via MDT. When fusion is blocked, MDT can still normally occur, but the nucleoids transported by MDT would not be possible to exchange among different mitochondria, thus leading to a large fraction of mitochondria devoid of nucleoids

(Supplementary Fig. 4). Therefore, mitochondrial fission, dynamic tubulation, and fusion may act in turn to facilitate the proper distribution in the mitochondrial network. In Fig 5 c & d, we show cases that nucleoids were transported to the tubular tip, and subsequent fusion of these tubules transferred the nucleoids from the donor mitochondrion to the acceptor mitochondrion. In our previous work, we observed that many dynamic tubules fuse with other mitochondria to form a membrane bridge between two mitochondria, which become part of the mitochondrial network. The percentage of fusion events mediated by dynamic tubulation is about 29.05%¹⁸.

Minor point: the authors should indicate in the figures what sort of treatments are being tested. Without that, it is difficult to follow.

Author reply:

We thank the reviewer for pointing out the problem. We have checked through the figure legends and added missing information of treatments including RNAi, immunolabeling etc..

Reference:

1. Chen XJ, Butow RA. The organization and inheritance of the mitochondrial genome. *Nat Rev Genet* **6**, 815-825 (2005).
2. Alam TI, *et al.* Human mitochondrial DNA is packaged with TFAM. *Nucleic Acids Res* **31**, 1640-1645 (2003).
3. Kukat C, Wurm CA, Spahr H, Falkenberg M, Larsson NG, Jakobs S. Super-resolution microscopy reveals that mammalian mitochondrial nucleoids have a uniform size and frequently contain a single copy of mtDNA. *Proc Natl Acad Sci U S A* **108**, 13534-13539 (2011).
4. Wang Y, Bogenhagen DF. Human mitochondrial DNA nucleoids are linked to protein folding machinery and metabolic enzymes at the mitochondrial inner membrane. *The J. Biol. Chem.* **281**, 25791-25802 (2006).
5. Yang R-F, *et al.* Suppression of Mic60 compromises mitochondrial transcription and oxidative phosphorylation. *Sci Rep* **5**, 7990-7999 (2015).
6. Modi S, *et al.* Miro clusters regulate ER-mitochondria contact sites and link cristae organization to the mitochondrial transport machinery. *Nat Commun* **10**, 4399 (2019).
7. Tsai PI, Papakyrikos AM, Hsieh CH, Wang X. Drosophila MIC60/mitofilin conducts dual roles in mitochondrial motility and crista structure. *Mol Biol Cell* **28**, 3471-3479 (2017).
8. Fransson S, Ruusala A, Aspenstrom P. The atypical Rho GTPases Miro-1 and Miro-2 have essential roles in mitochondrial trafficking. *Biochim Biophys Res Commun* **344**, 500-510 (2006).
9. Gieffers C, Koriath F, Heimann P, Ungermann C, Frey J. Mitofilin is a transmembrane protein of the inner mitochondrial membrane expressed as two isoforms. *Exp Cell Res* **232**, 395-399 (1997).
10. Kozjak-Pavlovic V. The MICOS complex of human mitochondria. *Cell Tissue Res* **367**, 83-93 (2017).
11. Misko A, Jiang S, Wegorzewska I, Milbrandt J, Baloh RH. Mitofusin 2 is necessary for transport of axonal mitochondria and interacts with the Miro/Milton complex. *J Neurosci* **30**, 4232-4240 (2010).
12. de Brito OM, Scorrano L. Mitofusin 2 tethers endoplasmic reticulum to mitochondria. *Nature* **456**, 605-610 (2008).
13. Naon D, *et al.* Critical reappraisal confirms that Mitofusin 2 is an endoplasmic reticulum-mitochondria tether. *Proc Natl Acad Sci U S A* **113**, 11249-11254 (2016).
14. Filadi R, Theurey P, Pizzo P. The endoplasmic reticulum-mitochondria coupling in health and disease: Molecules, functions and significance. *Cell Calcium* **62**, 1-15 (2017).
15. Friedman JR, Lackner LL, West M, DiBenedetto JR, Nunnari J, Voeltz GK. ER tubules mark sites of mitochondrial division. *Science* **334**, 358-362 (2011).
16. Lewis SC, Uchiyama LF, Nunnari J. ER-mitochondria contacts couple mtDNA synthesis with mitochondrial division in human cells. *Science* **353**, aaf5549 (2016).
17. Li H, *et al.* Mic60/Mitofilin determines MICOS assembly essential for mitochondrial dynamics and mtDNA nucleoid organization. *Cell Death Differ* **23**, 380-392 (2016).
18. Wang C, *et al.* Dynamic tubulation of mitochondria drives mitochondrial network formation. *Cell Res* **25**, 1108-1120 (2015).
19. Hessenberger M, *et al.* Regulated membrane remodeling by Mic60 controls formation of mitochondrial crista junctions. *Nat Commun* **8**, 15258 (2017).
20. Murley A, Nunnari J. The Emerging Network of Mitochondria-Organelle Contacts. *Mol Cell* **61**, 648-653 (2016).
21. Kornmann B, Osman C, Walter P. The conserved GTPase Gem1 regulates endoplasmic reticulum-mitochondria connections. *Proc Natl Acad Sci U S A* **108**, 14151-14156 (2011).
22. Vincent AE, Turnbull DM, Eisner V, Hajnóczky G, Picard M. Mitochondrial Nanotunnels. *Trends Cell Biol* **27**, 787799 (2017).
23. Vance JE. MAM (mitochondria-associated membranes) in mammalian cells: lipids and beyond. *Biochim Biophys Acta* **1841**, 595-609 (2014).

Reviewer comments, second version:

Reviewer #1 (Remarks to the Author):

The authors have done a thorough job in addressing this reviewer's comments. The new experiments add significant mechanistic insights. I recommend publication of the manuscript.

Reviewer #2 (Remarks to the Author):

The revised manuscript quality has been improved, and most of my concerns were addressed and resolved. But the quality of some Western blots is still poor, such as supplementary Fig. 9e; and supplementary Fig. 4b, 4e are over adjusted, the background are too white.

Reviewer #3 (Remarks to the Author):

The resubmission of Qin et al. has some substantial improvements. I appreciate their reexamination of thin tubules (MDTs) and normal mitochondrial branching, as well as the observation that some of the thin tubules can widen to become normal branches. Together with an analysis of nucleoid frequencies in the different classes of tubules strengthens the paper. The transfer of nucleoids through MDT is in my mind the most interesting part of this paper. It could have broader implications for redistribution of mtDNAs, for example in heteroplasmic cells. I am still not convinced by the arguments that Mic60 directly interacts with TFAM or Miro1. The amounts of coIP are very low, and no negative controls are shown. Some small amount of coIP is not surprising, considering the likelihood that these proteins are incorporated in several large protein complexes, but this can not be the whole story since there are many more cristae junctions with Micos than nucleoids or contacts with ER or coupling to microtubule transport. Other than that, I think the paper is fine.

Author rebuttal, second version:

Reviewer #1 (Remarks to the Author):

The authors have done a thorough job in addressing this reviewer's comments. The new experiments add significant mechanistic insights. I recommend publication of the manuscript.

Author reply:

Thanks for your comments and suggestions which helped to improve the manuscript significantly.

Reviewer #2 (Remarks to the Author):

The revised manuscript quality has been improved, and most of my concerns were addressed and resolved. But the quality of some Western blots is still poor, such as supplementary Fig. 9e; and supplementary Fig. 4b, 4e are over adjusted, the background are too white.

Author reply:

Thanks for your comments and suggestions which helped to improve the manuscript significantly. Regarding the Western blots, we have adjusted the contrast in supplementary Fig. 4b, 4e to make the background not too white.

Reviewer #3 (Remarks to the Author):

The resubmission of Qin et al. has some substantial improvements. I appreciate their reexamination of thin tubules (MDTs) and normal mitochondrial branching, as well as the observation that some of the thin tubules can widen to become normal branches. Together with an analysis of nucleoid frequencies in the different classes of tubules strengthens the paper. The transfer of nucleoids through MDT is in my mind the most interesting part of this paper. It could have broader implications for redistribution of mtDNAs, for example in heteroplasmic cells. I am still not convinced by the arguments that Mic60 directly interacts with TFAM or Miro1. The amounts of coIP are very low, and no negative controls are shown. Some small amount of coIP is not surprising, considering the likelihood that these proteins are incorporated in several large protein complexes, but this can not be the whole story since there are many more cristae junctions with Micos than nucleoids or contacts with ER or coupling to microtubule transport. Other than that, I think the paper is fine.

Author reply:

Thanks for your comments and suggestions which helped to improve the manuscript significantly. Regarding the question on the Co-IP data and the statement about the interaction between Mic60 and TFAM as well as the interaction between Mic60 and Miro1, we agree with the reviewer that the data would not be sufficient to support

“direct” protein-protein interaction, especially as Mic60 exists as a subunit of the MICOS complex. Therefore, in our revised manuscript, we did not state “direct interaction” based on our Co-IP data. Instead, we only suggested association between the protein molecules. Following the reviewer’s comment, we have toned down our conclusion on this part accordingly.